# FreeInv: Free Lunch for Improving DDIM Inversion

**Yuxiang Bao**[1,2*]**, Huijie Liu**[1*]**, Xun Gao**[2]**, Huan Fu**[2]**, Guoliang Kang**[1†]
[1] Beihang University, [2] HUJING Digital Media & Entertainment Group
{bbao5802, liuhuijie6410, kgl.prml}@gmail.com

## Abstract

Naive DDIM inversion process usually suffers from a trajectory deviation issue, i.e., the latent trajectory during reconstruction deviates from the one during inversion. To alleviate this issue, previous methods either learn to mitigate the deviation or design a cumbersome compensation strategy to reduce the mismatch error, exhibiting substantial time and computation cost. In this work, we present a nearly free-lunch method (named FreeInv) to address the issue more effectively and efficiently. In FreeInv, we randomly transform the latent representation and keep the transformation the same between the corresponding inversion and reconstruction time-step. It is motivated from a statistical perspective that an ensemble of DDIM inversion processes for multiple trajectories yields a smaller trajectory mismatch error on expectation. Moreover, through theoretical analysis and empirical study, we show that FreeInv performs an efficient ensemble of multiple trajectories. FreeInv can be freely integrated into existing inversion-based image and video editing techniques. Especially for inverting video sequences, it brings more significant fidelity and efficiency improvements. Comprehensive quantitative and qualitative evaluation on PIE benchmark and DAVIS dataset shows that FreeInv remarkably outperforms conventional DDIM inversion, and is competitive among previous state-of-the-art inversion methods, with superior computation efficiency.

## 1 Introduction

The recent developments of large-scale text-guided diffusion models, *e.g.* Stable Diffusion [36], have fueled the rise of image and video editing. To ensure the editing results are faithful to the original input, these methods typically employ Denoising Diffusion Implicit Models (DDIM) inversion [38] techniques. The DDIM inversion process involves mapping an image back to its noisy latent representation, from which the original image is expected to be reconstructed with high fidelity.

However, the reconstruction process usually suffers from a trajectory deviation issue, *i.e.*, the latent trajectory of the reconstruction process deviates from that of the inversion process, which means the error of latent representations between inversion and reconstruction may be accumulated along the denoising steps. This is because the ideal DDIM inversion and reconstruction process is theoretically based on the local linear assumption, *i.e.*, $\epsilon_\theta(x_t) \approx \epsilon_\theta(x_{t+1})$, where $\epsilon_\theta(\cdot)$ denotes the noise predicted by a neural network parameterized with $\theta$. The assumption usually does not hold in practice. Thus, the error introduced in each step will be accumulated and lead to a non-negligible deviation between the inversion and reconstruction trajectory, hampering the quality of reconstructed and edited results.

To mitigate the trajectory deviation, previous works focus on reducing the mismatch error of latent representations between inversion and reconstruction processes, *i.e.*, reducing $|\epsilon_\theta(x_t) - \epsilon_\theta(x_{t+1})|$ in each time-step. Learning-based methods [25, 5] aim to minimize the mismatch error through back-propagating the gradients to the null-text embedding (see Fig. 1(a)). Another group of methods [17,

---

*Equal contribution. †Corresponding author. Code is available at `https://github.com/yuxiangbao/FreeInv`. Project page is available at `https://yuxiangbao.github.io/FreeInv/`.

39th Conference on Neural Information Processing Systems (NeurIPS 2025).

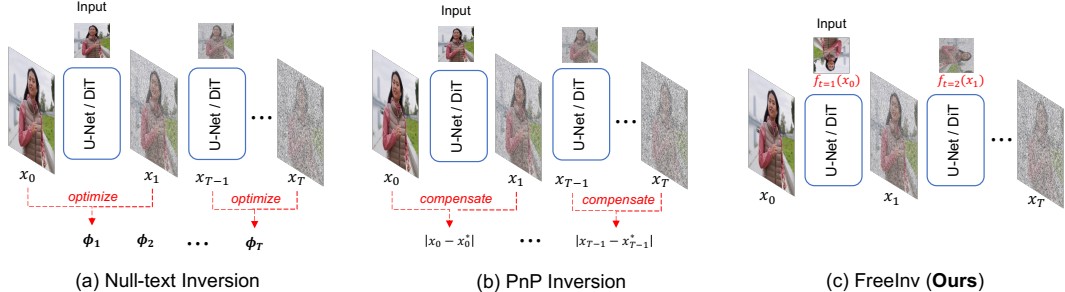

Figure 1: Illustration of different ways to mitigate the trajectory deviation in DDIM inversion. The small image above the networks denotes the input. (a) Null-text Inversion [25] reduces the mismatch error via optimizing a null-text embedding. (b) PnP Inversion [17] saves the reconstruction error of each step in memory and makes a compensation during the reconstruction or editing process. (c) FreeInv improves the DDIM inversion by applying random transformation (*e.g.* rotation) to the input latent, with negligible time or memory costs.

50, 15, 6] memorize or store the errors generated in each time-step, and exploit them to compensate for the latent representation deviation in the reconstruction procedure (see Fig. 1(b)). Although these techniques improve the reconstruction fidelity, they introduce high computation (time and memory) costs. Especially when inverting videos with hundreds of frames, they are cumbersome and inefficient.

In this paper, we propose a new method named *FreeInv* to deal with the trajectory deviation issue in a nearly free-lunch manner. Specifically, from a statistical perspective, we find that an ensemble of the inversion and the reconstruction processes for multiple image samples yields a smaller mismatch error. In detail, we average the predicted noise from multiple images for inversion and reconstruction and find the mismatch error can be suppressed. Further, based on our theoretical analysis and empirical observations, we propose FreeInv, which is a simplified version of performing trajectory ensemble. As illustrated in Fig. 1(c), in FreeInv, we randomly transform (*e.g.,* rotate) the latent representation and keep the transformation (*e.g.,* rotation angles) the same between the corresponding inversion and reconstruction time-step. Thanks to operational simplicity, FreeInv can be readily plugged into U-Net [13, 36] and DiT [29, 1, 7] architectures. Extensive experiments on PIE benchmark [17] demonstrate that FreeInv significantly outperforms the DDIM baseline, and achieves performance comparable or superior to existing state-of-the-art inversion approaches [25, 45, 46, 50, 17, 15, 9, 47]. For efficiency, our method introduces negligible costs compared to the DDIM baseline and consumes much smaller time and memory than all previous inversion methods tailored for mitigating the trajectory deviation. Due to the efficient design, FreeInv is well-suited for the inversion of video sequences. When combined with TokenFlow [10], FreeInv exhibits superior reconstruction/editing fidelity and efficiency, compared to previous state-of-the-art inversion method STEM-Inv [20].

In a nutshell, our contribution is summarized as follows

- From the statistical perspective, we find that an ensemble of trajectories for multiple images can effectively reduce the latent mismatch error between inversion and reconstruction processes, thereby improving reconstruction fidelity effectively.

- We propose a method named FreeInv to perform an efficient ensemble of trajectories, *i.e.,* we randomly transform (*e.g.,* rotate) the latent representations and keep the transformation (*e.g.,* rotation angles) at each step the same between the inversion and reconstruction processes.

- FreeInv is compatible with both U-Net and DiT architectures. Its efficient design enables it to be applied not only to image reconstruction but also to video sequences. Extensive experiments demonstrate that FreeInv achieves reconstruction performance on par with, or even exceeding, existing inversion methods, while offering significantly improved efficiency.

## 2 Related Works

**Text guided image editing.** Recently, text-guided diffusion models [26, 37, 36, 13, 38, 39, 40] offer significantly more powerful and flexible image editing capabilities compared to previous

methods [16, 53, 27, 28, 44, 48, 22, 35, 33, 54, 18]. Text-guided image editing requires editing the image following the prompt while maintaining the main components of the original image. Imagic [19] and UniTune [43] achieve this goal through restrictive fine-tuning of the pretrained model. Blended diffusion [2] and GLIDE [26] utilize the provided mask to control the editing region, which is not user-friendly. To realize controllable and precise editing with prompt, Prompt-to-Prompt [12] introduces the feature and attention map from the DDIM reconstruction process into the editing process, achieving promising editing results.

**DDIM inversion for image/video editing.** In image/video editing tasks, DDIM inversion technique is widely adopted [12, 4, 42, 10, 32, 23, 3, 14, 51, 8]. However, the editing results are always constrained by the reconstruction quality. Naive DDIM inversion usually suffers from a trajectory deviation issue, leading to distorted reconstruction. A lot of works [25, 5, 24, 21, 45, 15, 50, 17, 46] have been proposed to alleviate this issue. Null-text Inversion [25] minimizes the reconstruction error at each time-step through optimizing the null-text embedding. Other works [17, 50, 6] choose to save the error with extra memory occupation and make compensation during editing process. EDICT [45] designs a parallel inversion and reconstruction process to realize accurate preservation. Although these works are well applied in image inversion, but few of them are suited for processing video sequences for the increased computation burden. STEM Inversion [20] is designed to invert video sequences, but it still requires iterations to calculate a compact video representation. In comparison, FreeInv offers a free-lunch and more general alternative for both image and video inversion.

## 3 Methodology

### 3.1 DDIM Inversion Revisiting

Recently, Song et al. [38] proposed the Denoising Diffusion Implicit Model (DDIM), which serves as an efficient technique for diffusion model sampling, following the formula

$$\frac{x_t}{\sqrt{\alpha_t}} = \frac{x_{t+1}}{\sqrt{\alpha_{t+1}}} - \eta_t \cdot \epsilon_\theta \left( x_{t+1}, t+1 \right),$$

$$\eta_t = \sqrt{\frac{1 - \alpha_{t+1}}{\alpha_{t+1}}} - \sqrt{\frac{1 - \alpha_t}{\alpha_t}}, \tag{1}$$

where $x_t$ denotes the latent at time-step $t$, and $\epsilon_\theta \left( \cdot, \cdot \right)$ refers to the noise prediction network with parameter $\theta$. Note that there are no stochastic terms in the formula, meaning that the sampling procedure is deterministic. Therefore, if we inverse the DDIM sampling process from $t = 0$ to $t = T$, where $T$ denotes the total sampling steps, we can get the initial noisy latent of the original image. This inversion process can be formulated as

$$\frac{x_{t+1}}{\sqrt{\alpha_{t+1}}} = \frac{x_t}{\sqrt{\alpha_t}} + \eta_t \cdot \epsilon_\theta \left( x_t, t+1 \right). \tag{2}$$

If performing DDIM sampling on the inverted noisy latent can recover the original image ideally, it may greatly benefit diffusion-based image/video editing tasks [9, 17], which aims to modify part of the image while keeping the rest unchanged.

However, due to the discrete nature, slight error exists in each reconstruction time-step, resulting in flawed reconstruction. In order to quantitatively describe the reconstruction error, let us consider the adjacent time-step $t$ and $t + 1$, where the latent is inverted from time-step $t$ to $t + 1$ and then goes back to $t$. The error can be formulated as

$$|x_t^* - x_t| = \sqrt{\alpha_t} \eta_t \cdot |\epsilon_\theta \left( x_{t+1}, t+1 \right) - \epsilon_\theta \left( x_t, t+1 \right)|, \tag{3}$$

where $| \cdot |$ refers to calculating element-wise absolute value and we utilize $x_t^*$ and $x_t$ to represent the reconstructed latent and the inverted latent, respectively. Because $\alpha_t$ is the hyper-parameter predefined in DDIM schedule that keeps unchanged, the reconstruction error is determined by the mismatch error $|\epsilon_\theta \left( x_{t+1}, t+1 \right) - \epsilon_\theta \left( x_t, t+1 \right)|$. The ideal DDIM inversion and reconstruction process assumes that $\epsilon_\theta \left( x_{t+1}, t+1 \right) \approx \epsilon_\theta \left( x_t, t+1 \right)$. Such an error will be accumulated along the time-steps and become non-negligible. As a result, the trajectory of the reconstruction deviates from the one of the inversion process, thus hampering the fidelity of reconstruction results.

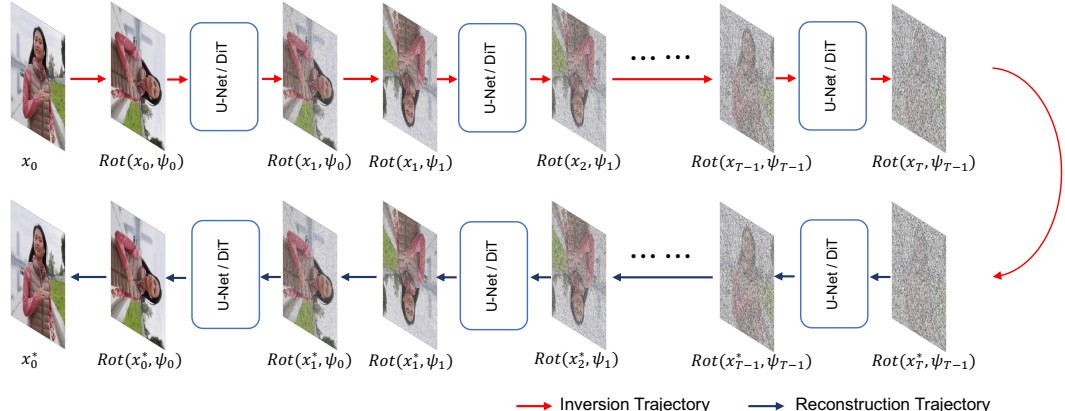

→ Inversion Trajectory    → Reconstruction Trajectory

Figure 2: Detailed illustration of FreeInv. We employ rotation $Rot(\cdot, \cdot)$ as the transformation $f(\cdot)$ for example. During both the inversion and reconstruction phases, we rotate the latent representation with the same angle $\psi_t$ at the $t$-th time-step, where $\psi_t$ is randomly sampled.

## 3.2  Multi-Branch DDIM Inversion

As discussed in Sec. 3.1, the key of high-fidelity DDIM inversion is to minimize the mismatch error in each time-step. Inspired by the ensemble techniques [11] in various computer vision tasks, we propose to ensemble multiple trajectories to enhance reconstruction fidelity by constructing a multi-branch DDIM inversion (MBDI) and reconstruction.

Specifically, when inverting one image, an arbitrary number of different images are also sampled as auxiliary samples and follow the parallel inversion and reconstruction trajectory. In each time-step, instead of inverting/reconstructing each branch independently, we make an ensemble of the noise predictions from all the branches to invert/reconstruct all the branches simultaneously. Specifically, we average all the noise predictions at each time-step, which is

$$\epsilon_{\theta,\tilde{\boldsymbol{\lambda}}}^{e}\left(x_t, t+1\right) = \sum_{i=1}^{N} \tilde{\lambda}_i \epsilon_\theta\left(x_t^i, t+1\right), \tag{4}$$

where $\tilde{\boldsymbol{\lambda}} = [\tilde{\lambda}_1, \tilde{\lambda}_2, \cdots, \tilde{\lambda}_N] = \left[\frac{1}{N}, \frac{1}{N}, \cdots, \frac{1}{N}\right]$. In Eq. (4), $x_t^i$ refers to the latent of the $i$-th branch at time-step $t$. For reconstruction, the ensemble noise $\epsilon_{\theta,\tilde{\boldsymbol{\lambda}}}^{e}\left(x_{t+1}, t+1\right)$ can be obtained in a similar way. Compared with Eq. (1) and (2), the latent in each branch is inverted and reconstructed with $\epsilon_{\theta,\tilde{\boldsymbol{\lambda}}}^{e}\left(x_t, t+1\right)$ and $\epsilon_{\theta,\tilde{\boldsymbol{\lambda}}}^{e}\left(x_{t+1}, t+1\right)$ instead of $\epsilon_\theta\left(x_t, t+1\right)$ and $\epsilon_\theta\left(x_{t+1}, t+1\right)$ respectively. Note that this modification does not affect the deterministic nature of the procedure in the sense that the original image can still be reconstructed theoretically.

**MBDI reduces mismatch error.**  With MBDI, the mismatch error between estimated noise in inversion and reconstruction is

$$|\epsilon_{\theta,\tilde{\boldsymbol{\lambda}}}^{e}\left(x_{t+1}, t+1\right) - \epsilon_{\theta,\tilde{\boldsymbol{\lambda}}}^{e}\left(x_t, t+1\right)| = \frac{1}{N} | \sum_{i=1}^{N} \left[\epsilon_\theta\left(x_{t+1}^i, t+1\right) - \epsilon_\theta\left(x_t^i, t+1\right)\right] |. \tag{5}$$

According to the triangle inequality we get

$$|\epsilon_{\theta,\tilde{\boldsymbol{\lambda}}}^{e}\left(x_{t+1}, t+1\right) - \epsilon_{\theta,\tilde{\boldsymbol{\lambda}}}^{e}\left(x_t, t+1\right)| \leq \frac{1}{N} \sum_{i=1}^{N} |\epsilon_\theta\left(x_{t+1}^i, t+1\right) - \epsilon_\theta\left(x_t^i, t+1\right)|. \tag{6}$$

The right side of the inequality is the mean error of each independent branch. Given that the initial samples $x_0^i$ are independently drawn from the natural image distribution and the inversion procedure is deterministic, the right side of Eq. 6 can be considered as an unbiased estimation for the expectation of mismatch error $\mathbb{E}_{x_0 \sim p(x_0)}\{|\epsilon_\theta\left(x_{t+1}, t+1\right) - \epsilon_\theta\left(x_t, t+1\right)|\}$. Given that the reconstruction error is proportional to the mismatch error in Eq. (3), the reconstruction error of multi-branch is no larger than that of a single branch in each time-step on expectation. Therefore, performing the ensemble of multiple trajectories may yield better reconstruction results compared to a single branch.

### 3.3 Free-lunch DDIM Inversion

Though effective, the computation and memory cost of $N$-branch inversion framework is high, and the cost is approximately $N$ times more than the standard DDIM inversion. Thus, in FreeInv, we make two modifications to improve the efficiency.

**(1) One-time MC sampling at each time-step.** Different from deterministic $\tilde{\boldsymbol{\lambda}}$ in MBDI in Eq. (4), we introduce a random variable $\boldsymbol{\lambda}^t = [\lambda_1^t, \lambda_2^t, \cdots, \lambda_N^t] \sim \text{Categorical}\left(\frac{1}{N}, \frac{1}{N}, \ldots, \frac{1}{N}\right)$ (in the following, we omit superscript $t$ which denotes the time-step for simplicity). Then, we obtain $\epsilon_{\theta,\boldsymbol{\lambda}}^e(x_t, t+1) = \sum_{i=1}^N \lambda_i \epsilon_\theta(x_t^i, t+1)$. The expectation of $\epsilon_{\theta,\boldsymbol{\lambda}}^e(x_t, t+1)$ over $\boldsymbol{\lambda}$ can be denoted as

$$\mathbb{E}_{\boldsymbol{\lambda}}[\epsilon_{\theta,\boldsymbol{\lambda}}^e(x_t, t+1)] = \frac{1}{N}\sum_{i=1}^N \epsilon_\theta(x_t^i, t+1),\tag{7}$$

which means $\mathbb{E}_{\boldsymbol{\lambda}}[\epsilon_{\theta,\boldsymbol{\lambda}}^e(x_t, t+1)]$ equals to performing MBDI. Therefore, the key is to estimate $\mathbb{E}_{\boldsymbol{\lambda}}[\epsilon_{\theta,\boldsymbol{\lambda}}^e(x_t, t+1)]$. We utilize Monte Carlo (MC) sampling to estimate $\mathbb{E}_{\boldsymbol{\lambda}}[\epsilon_{\theta,\boldsymbol{\lambda}}^e(x_t, t+1)]$. For efficiency reasons, we only perform one-time MC sampling at each inversion time-step, *i.e.,*

$$\epsilon_{\theta,\hat{\boldsymbol{\lambda}}}^e(x_t, t+1) = \sum_{i=1}^N \hat{\lambda}_i \epsilon_\theta(x_t^i, t+1),\tag{8}$$

where $\hat{\boldsymbol{\lambda}} = [\hat{\lambda}_1, \hat{\lambda}_2, \cdots, \hat{\lambda}_N]$ is a one-hot vector sampled from the distribution of $\boldsymbol{\lambda}$. Thus, only one branch is randomly sampled at each time-step of inversion. Note that MC sampling is performed independently at each time-step in FreeInv, essentially distinguishing it from single-branch DDIM inversion which can be treated as deterministically selecting one branch at all time-steps. We empirically (Sec. 4.4) find that it performs comparably to multi-time MC sampling and MBDI.

**(2) Image transformation as a branch.** Through one-time MC estimation, only one branch is randomly selected at each time-step to estimate the noise instead of using all $N$ branches, effectively reducing time consumption. However, the memory cost remains high, as it is still needed to maintain the latent representations of all $N$ branches.

To further improve efficiency, we replace the explicit multiple branches by applying transformations (*e.g.* rotation, flipping, patch-shuffling, *etc.*) to the image/latent representation, thereby generating multiple augmented versions. Since FreeInv does not impose any spatial or semantic constraints on different branches, it is reasonable to mimic multi-branch image sampling through transformation, *i.e.* $x_t^i = f_i(x_t)$, where $f_i(\cdot)$ refers to the transformation that implicitly represents the $i$-th branch. Then Eq. (8) becomes:

$$\epsilon_{\theta,\hat{\boldsymbol{\lambda}}}^f(x_t, t+1) = \sum_{i=1}^N \hat{\lambda}_i \epsilon_\theta(f_i(x_t), t+1).\tag{9}$$

Overall, following Eq. (1-2), the inversion and reconstruction process of FreeInv can be formulated as

$$\frac{x_{t+1}}{\sqrt{\alpha_{t+1}}} = \frac{x_t}{\sqrt{\alpha_t}} + \eta_t \cdot \epsilon_{\theta,\hat{\boldsymbol{\lambda}}}^f(x_t, t+1),\tag{10}$$

$$\frac{x_t}{\sqrt{\alpha_t}} = \frac{x_{t+1}}{\sqrt{\alpha_{t+1}}} - \eta_t \cdot \epsilon_{\theta,\hat{\boldsymbol{\lambda}}}^f(x_{t+1}, t+1).\tag{11}$$

In Fig. 2, we employ rotation operation as the transformation for example to illustrate the whole process. In detail, during the inversion process, for an image, we rotate the latent code $x_t$ at each time-step $t$ with a randomly selected angle (*i.e.,* $0$, $\pi/2$, $\pi$, $3\pi/2$) to predict the noise, implicitly formulating a 4-branch DDIM inversion process. For the reconstruction process, we apply similar operation. To ensure consistency, we keep the rotation angle the same between the inversion and reconstruction processes at each time-step.

In this way, the additional computational consumption is limited to the transformation and the memory required to store the transformation type (*e.g.* rotation angles), both of which are negligible compared to previous methods tailored for mitigating the trajectory deviation (see Sec. 4.2 for computation cost comparisons).

Table 1: **Quantitative comparison: reconstruction.** We quantitatively evaluate reconstruction faithfulness, as well as computation costs of existing inversion methods, including U-Net based and DiT based methods, on the PIE benchmark. FreeInv achieves competitive results with superior high efficiency. All the U-Net based methods use Stable Diffusion 1.5 and 50-step schedule except VI uses the Latent Consistency Model (LCM) and 12-step schedule. All the DiT based methods adopt 25-step schedule.

| Methods | | Reconstruction Accuracy | | | | Inversion Computation Costs | |
| | | PSNR $\uparrow$ | LPIPS $(\times 10^{-2}) \downarrow$ | MSE $(\times 10^{-3}) \downarrow$ | SSIM $\uparrow$ | Time (Seconds) $\downarrow$ | Memory (MB) $\downarrow$ |
|---|---|---|---|---|---|---|---|
| U-Net Based | DDIM Baseline [38] | 25.04 | 9.14 | 4.43 | 0.77 | 4 | **3031** |
| | NTI [25] | 26.74 | 5.46 | 3.13 | 0.79 | 148 | 11945 |
| | EDICT [45] | 27.21 | 5.12 | 2.88 | 0.80 | 81 | 12325 |
| | DI [15] | **28.19** | **4.76** | **2.29** | **0.81** | 16 | 13595 |
| | VI [50] | 27.86 | 5.45 | 3.77 | 0.80 | 3 | 13853 |
| | ReNoise [9] | 26.61 | 6.52 | 3.19 | 0.79 | 21 | 6395 |
| | BELM [46] | 27.12 | 5.15 | 2.91 | 0.79 | 5 | 3641 |
| | PI [17] | 27.12 | 5.13 | 2.91 | 0.79 | 4 | 7197 |
| | **Ours** | 27.69 | 5.14 | 2.45 | **0.81** | 4 | **3031** |
| DiT Based | FLUX [1] | 14.92 | 38.60 | 46.19 | 0.54 | 7 | **32430** |
| | FLUX+RF-Solver [47] | 26.38 | 10.98 | 3.89 | 0.84 | 15 | **32430** |
| | FLUX+**Ours** | **29.24** | **4.25** | **1.64** | **0.90** | 7 | **32430** |

Table 2: **Quantitative comparison: editing.** With P2P as baseline, we quantitatively compare existing inversion methods, with regard to background preservation and description alignment of edited images.

| Method | | Structure Distance $(\times 10^{-3}) \downarrow$ | Background Preservation | | | | CLIP Similarity | |
| Inversion | Editing | | PSNR $\uparrow$ | LPIPS $(\times 10^{-2}) \downarrow$ | MSE $(\times 10^{-3}) \downarrow$ | SSIM $\uparrow$ | Whole $\uparrow$ | Edited $\uparrow$ |
|---|---|---|---|---|---|---|---|---|
| DDIM [38] | P2P | 69.88 | 17.84 | 21.02 | 22.07 | 0.71 | 25.18 | 22.33 |
| NTI [25] | P2P | 10.11 | 27.80 | 4.99 | 2.99 | 0.85 | 24.80 | 21.76 |
| EDICT [45] | P2P | **3.84** | **29.79** | **3.70** | **2.04** | **0.87** | 23.09 | 20.32 |
| DI [15] | P2P | 11.64 | 25.96 | 6.16 | 3.93 | 0.84 | **25.60** | **22.61** |
| VI [50] | P2P | 17.35 | 28.00 | 5.75 | 7.61 | 0.85 | 24.86 | 22.12 |
| ReNoise [9] | P2P | 23.25 | 25.11 | 8.97 | 5.14 | 0.82 | 23.81 | 21.16 |
| BELM [46] | P2P | 17.28 | 25.51 | 8.46 | 4.76 | 0.82 | 24.23 | 21.30 |
| PI [17] | P2P | 10.89 | 27.21 | 5.44 | 3.31 | 0.85 | 25.02 | 22.12 |
| **Ours** | P2P | 17.13 | 26.03 | 6.79 | 4.17 | 0.83 | 25.30 | 22.33 |

# 4 Experiments

We make both quantitative and qualitative comparison with state-of-the-art inversion enhancing techniques, covering Null-Text Inversion (NTI) [25], EDICT [45], DDPM Inversion (DI) [15], Virtual Inversion (VI) [50], PnP Inversion (PI) [17], ReNoise [9], BELM [46] and STEM Inversion [20]. Moreover, we conduct comprehensive experiments by plugging FreeInv into popular inversion-based image/video editing approaches, including Prompt-to-Prompt (P2P) [12], MasaCtrl [4], PnP [42], and TokenFlow [10]. We also conduct ablation studies to provide a more comprehensive understanding of our method.

## 4.1 Implementation Details

In our experiments, unless otherwise stated, we adopt Stable Diffusion [36] 1.5 with a 50-step DDIM schedule for U-Net based methods, and FLUX.1-dev [1] (abbr. FLUX) with a 25-step schedule for DiT based methods.

## 4.2 Image Reconstruction and Editing

**Dataset.** Following previous works [17, 50], we employ the PIE-benchmark [17] and its officially released code to quantitatively evaluate image editing results from FreeInv and the compared methods. PIE-benchmark consists of 700 images of resolution $512 \times 512$, the content of which is from nature or artificial generation. In the benchmark, each image is associated with a source prompt, an editing prompt, and an editing mask indicating anticipated editing areas.

**Evaluation Metrics.** For the image reconstruction task, we employ PSNR, LPIPS [52], MSE, and SSIM [49] to evaluate the reconstruction quality. In addition, we record the time cost and GPU memory usage to assess the computational efficiency. For the image editing task, we utilize structure

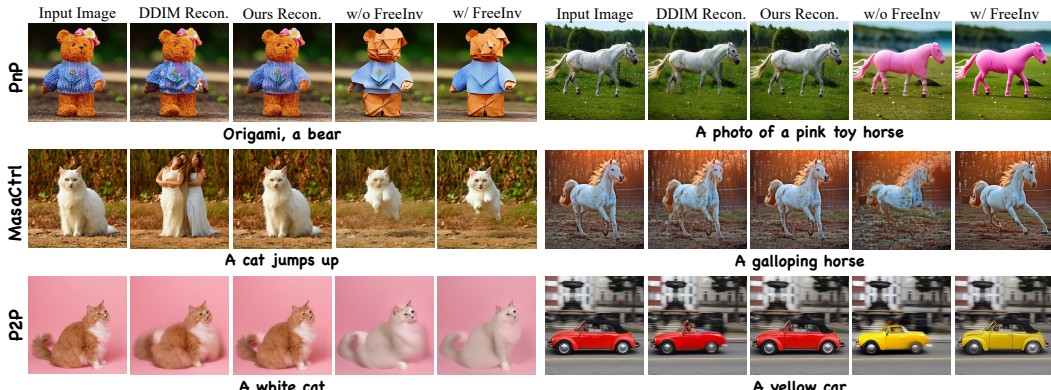

Figure 3: **Qualitative comparison.** We integrate FreeInv into PnP [42], MasaCtrl [4], and P2P [12], respectively. We compare the reconstruction and editing results w/ or w/o FreeInv.

distance [41] and background preservation metrics [17] to measure how well the layout and the unedited regions are preserved. Furthermore, CLIP similarity [34] is adopted to assess the alignment between the edited image and the target textual prompt.

**Quantitative Evaluation.** Quantitative results for image reconstruction are provided in Tab. 1. The results show that (i) **Effectiveness**: We observe that FreeInv and the other existing inversion methods boost the reconstruction accuracy effectively, where FreeInv achieves competitive or even superior results. (ii) **Generality**: FreeInv can be seamlessly integrated with U-Net based or DiT based methods, benefiting from its ultra simple design and implementation. (iii) **Efficiency**: Unlike other methods that rely on complicated numerical solvers (*e.g.* BELM, EDICT), require gradient back-propagation (*e.g.* NTI, ReNoise), or need extra memory consumption (*e.g.* DI, VI, PI), FreeInv incurs negligible computational overhead, *i.e.,* the computational cost of FreeInv is approximately equal to that of the DDIM baseline. Quantitative comparison for image editing is presented in Tab. 2, where we adopt P2P as the baseline editing framework, and all the inversion approaches are integrated into it. With inversion techniques, the edited images show improved faithfulness to the original content, with lower structure distance and better-preserved backgrounds. Compared to existing methods, FreeInv achieves superior prompt-image alignment, as indicated by its high CLIP similarity.

**Qualitative Comparison.** In Fig. 3, we visualize the reconstruction results and the corresponding editing outcomes of PnP [42], MasaCtrl [4], and P2P [12] with or without FreeInv. Due to the poor preservation of naive DDIM inversion, reconstruction results without FreeInv often exhibit significant deviations from the original image. In contrast, FreeInv significantly improves

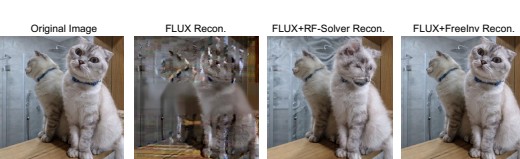

Figure 4: Visualization of the reconstructed images of different approaches with FLUX.

reconstruction quality. This improvement in reconstruction fidelity further leads to better editing outcomes, such as restoring the distorted face of the teddy bear and harmonizing the color of the horse's legs. Besides U-Net based methods, we also evaluate the effectiveness of FreeInv on DiT-based approaches. In Fig. 4, we present the reconstruction results of FLUX, FLUX+RF-Solver [47], and FLUX+FreeInv, where the results of FLUX+FreeInv demonstrate superior fidelity. More visualization results are shown in the appendix.

Moreover, Fig. 5 presents the editing results of P2P equipped with different diffusion inversion methods. P2P with naive DDIM inversion can hardly maintain the structure or details of the input image. In contrast, EDICT, DI, VI, and BELM exhibit such a strong fidelity to the source image that it hinders their editing capabilities. Overall, the editing results from NTI, PI, and FreeInv demonstrate strong alignment

Table 3: **Human evaluation.** We conduct a user study on the preference of editing results w/o or w/ FreeInv. The details about the user study are provided in the appendix.

| User Preference | PnP | MasaCtrl | P2P |
|---|---|---|---|
| w/o FreeInv | 18.18 | 12.73 | 10.34 |
| w/ FreeInv | **81.82** | **87.27** | **89.66** |

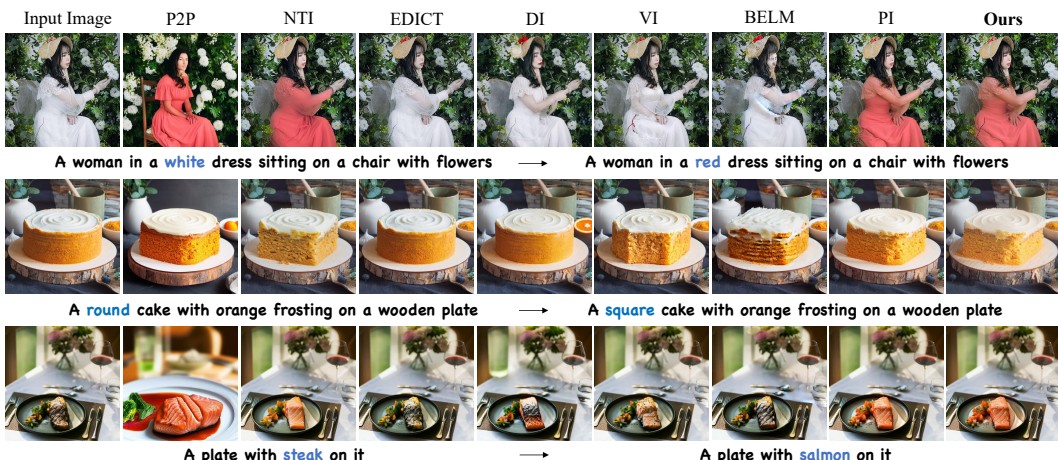

Input Image P2P NTI EDICT DI VI BELM PI **Ours**

A woman in a white dress sitting on a chair with flowers ⟶ A woman in a red dress sitting on a chair with flowers

A round cake with orange frosting on a wooden plate ⟶ A square cake with orange frosting on a wooden plate

A plate with steak on it ⟶ A plate with salmon on it

Figure 5: **Qualitative comparison.** We select Prompt-to-Prompt (P2P) as the baseline editing framework, and compare the editing results with different inversion approaches. The source and target prompts are provided below each row of the images.

with the target prompt, as well as high faithfulness with respect to the input. More comparisons can be found in the appendix.

**Human Evaluation.** To further validate the effectiveness of FreeInv, we also conduct a user study to calculate the user preference rate of the edited images with the editing instruction. The edited images generated with FreeInv achieve significantly higher user preference compared to those without FreeInv, as illustrated in Tab. 3. In the appendix, we additionally provide the human preference rate of the editing results with different inversion methods, as well as more implementation details about the user study.

### 4.3 Video Reconstruction and Editing

**Dataset and Metrics** Following previous works [10, 20], we use the videos from the DAVIS dataset [31] or downloaded from the Internet for evaluation. The video is captured for the first 120 frames, cropped to a resolution of $512 \times 512$ pixels. We measure the mean PSNR across all the frames to evaluate the reconstruction fidelity. Additionally, time and memory costs are reported to compare efficiency.

**Quantitative and Qualitative Comparison.** TokenFlow [10] is adopted as the baseline method, which is representative for inversion-based video editing. To demonstrate the superiority of FreeInv, it is also compared with STEM Inversion [20], a state-of-the-art method designed for video inversion. The quantitative and qualitative comparisons are presented in Fig. 6. More visualization results and videos that can be played are available in the appendix. Through Fig. 6, it can be seen that (i) DDIM inversion and reconstruction exhibit poor preservation of the original video contents. Consequently, in the pixar animation case, the eyes of the woman look strange, and in the black SUV case, the road appears dark at some regions. (ii) Although STEM Inversion makes a great improvement, there remains artifacts in the reconstruction with regard to some details, which are annotated with red boxes. Moreover, we notice the editing results with STEM-Inv are usually over sharpened, *e.g.*, the cloud in the pixar animation case and the trace on the road in the black SUV case. (iii) In comparison, FreeInv achieves the best reconstruction results regarding the highest PSNR value and visualization faithfulness, and brings negligible extra consumption (2MB GPU memory occupation). The enhanced reconstruction quality benefits editing, making editing results more natural and detailed.

### 4.4 Ablation Study

**MC sampling *vs*. MBDI.** We compare the reconstruction quality of MC sampling and MBDI. We adopt MBDI ($N = 4$) as the baseline and compare it with one-time, two-time, and four-time

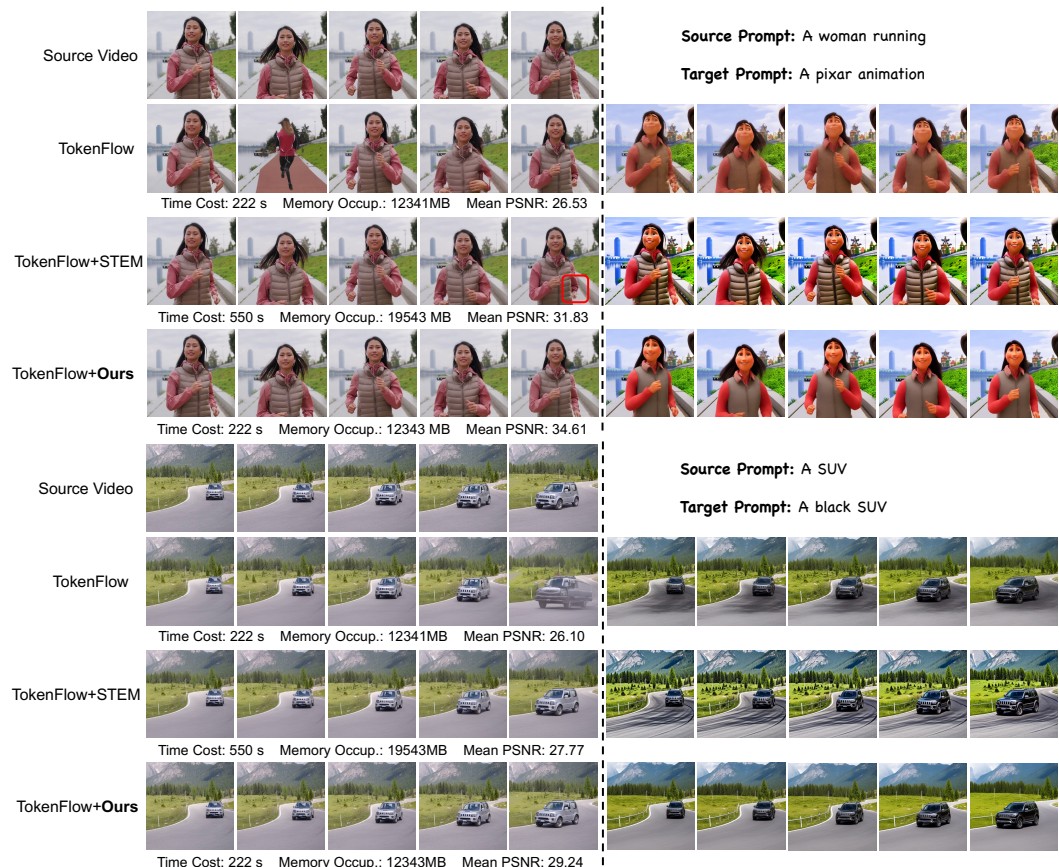

Figure 6: **Video comparison.** We compare TokenFlow [10], TokenFlow+STEM [20], and Token-Flow+Ours with respect to the reconstruction results (on the left side of the dash-line), the editing outcome (on the right side of the dash-line), as well as time and memory costs (below the reconstruction results).

MC sampling. The comparison is shown in Tab. 4. We observe that one-time sampling performs comparably to MC sampling with multiple times and MBDI.

Table 4: Ablation study on the number of MC sampling steps.

| Sampling Times | PSNR ↑ | LPIPS ($\times 10^{-2}$) ↓ | MSE ($\times 10^{-3}$) ↓ | SSIM ↑ |
|---|---|---|---|---|
| MBDI (4-branch) | **28.14** | **4.94** | **2.30** | **0.81** |
| 1-time MC (**Ours**) | 28.13 | 5.00 | **2.30** | **0.81** |
| 2-time MC | **28.14** | 5.00 | **2.30** | **0.81** |
| 4-time MC | **28.14** | 5.00 | **2.30** | **0.81** |

**Transformation *vs.* Multiple Images.** As discussed in Sec. 3.3, instead of sampling different images in multiple branches, we exploit different transformations to improve efficiency. For multi-branch DDIM inversion, we compare two variations. One is that each branch consists of distinct images, termed as **MB-I** in our experiment, while the other is that each branch consists of the original image rotated with different angles, termed as **MB-R**. In the ablation, the branch number is set to 4. The comparison is listed in Tab. 5. FreeInv performs comparable to **MB-I** and **MB-R**, but outperforms the DDIM baseline by a large margin (27.64 versus 25.04 in terms of PSNR).

**Comparison between different types of transformations.** We implement FreeInv with different types of transformations including random rotation, random horizontal/vertical flipping, random patch shuffling, random color jittering, as well as the combination of these transformations for comparison. The experiment is conducted on the PIE benchmark.The results in Tab. 6 show that different types of transformations achieve comparable performance in improving the reconstruction faithfulness.

Table 5: Comparison among a) **MB-I**: multi-branch inversion where each branch represents distinct images, b) **MB-R**: multi-branch inversion where each branch corresponds to one image rotated a certain angle, and c) **Ours**: single-branch inversion where random rotation is applied in each time-step.

| Methods | PSNR ↑ | LPIPS ($\times 10^{-2}$)↓ | MSE ($\times 10^{-3}$)↓ | SSIM ↑ |
|---------|--------|------------|-----------|--------|
| DDIM    | 25.04  | 9.14       | 4.43      | 0.77   |
| **MB-I**  | **28.14** | **4.94**   | **2.30**  | **0.81** |
| **MB-R**  | 27.73  | 5.06       | 2.42      | **0.81** |
| **Ours**  | 27.64  | 5.14       | 2.45      | **0.81** |

Table 6: Comparison between different types of transformations.

| Methods | PSNR ↑ | LPIPS ($\times 10^{-2}$)↓ | MSE ($\times 10^{-3}$)↓ | SSIM ↑ |
|---------|--------|------------|-----------|--------|
| flip          | 27.47 | 5.43   | 2.55   | 0.80 |
| patch shuffle | 27.61 | 5.18   | 2.55   | 0.80 |
| color jitter  | 27.53 | 5.40   | 2.55   | 0.80 |
| rotation      | 27.64 | **5.14** | **2.45** | **0.81** |
| combination   | **27.69** | **5.14** | **2.45** | **0.81** |

**Cross-attention Map Visualization** To provide an intuitive understanding of the improvement brought by FreeInv, we visualize the cross-attention map in Fig. 7. The prompt is "a woman running", and we aggregate the cross-attention maps with respect to the word "woman" among all time-steps for each sample. We observe that FreeInv enables the model to focus more precisely on the region of "woman" compared to the DDIM baseline.

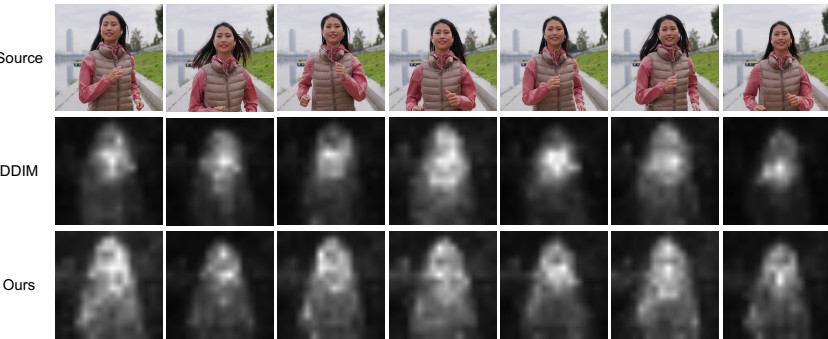

Figure 7: Visualization of cross-attention map in diffusion U-Net.

## 5   Conclusion

In this paper, we find that an ensemble of trajectories for multiple images can effectively reduce the DDIM reconstruction error. Based on such a finding, we propose a method named FreeInv to perform an efficient ensemble. FreeInv enhances DDIM inversion in a free-lunch manner. In detail, we randomly transform the latent representation, and keep the transformation at each time-step the same between the inversion and the reconstruction. FreeInv is compatible with both U-Net and DiT architectures. Thanks to its efficiency, FreeInv is applicable not only to image reconstruction but also to video sequences. In both image and video reconstruction tasks, it achieves reconstruction fidelity comparable to or better than existing methods, while demonstrating significantly improved efficiency.

**Acknowledgments**

This project is supported by National Natural Science Foundation of China under Grant 92370114. This work is also supported by HUJING Digital Media & Entertainment Group through HUJING Digital & Entertainment Innovative Research Program.

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

# Appendix

In the appendix, we provide more quantitative and qualitative results to facilitate a more comprehensive understanding and evaluation of the proposed method. Moreover, we supplement more visualization results comprising both image and video editing in a web-page through the link https://yuxiangbao.github.io/FreeInv/ for better visualization. Further, we discuss the social impact and limitations of FreeInv.

## A   Image/Video Editing with FreeInv

As FreeInv may improve the reconstruction quality in a free-lunch manner, it can be easily combined with existing image/video editing approaches [12, 42, 4, 10] to improve the editing performance. As current editing methods typically rely on spatial coherence (*e.g.,* the self-attention map in U-Net) as guidance, we make minor modifications of FreeInv to make it better aligned with existing editing approaches, *i.e.,* we additionally apply inverse transformation to the predicted noise. For example, at a certain time-step, we rotate the latent by an angle to predict the noise and then reversely rotate the noise with the same angle before adding the noise to the latent $x_t$. Note that such an inverse transformation of noise is optional for reconstruction as the reconstruction quality is mainly determined by the closeness between inversion and reconstruction trajectories. We compare the reconstruction and editing results with and without applying inverse transformation on the predicted noise on PIE benchmark. The results presented in Tab. 7 and Fig. 8 show that the inverse transformation has minimal impact on the reconstruction results, but may benefit the structural faithfulness in editing.

Table 7: Ablation study on the effect of inverse transformation applied on the predicted noise with respect to reconstruction quality on the PIE benchmark.

| Methods | PSNR ↑ | LPIPS ($\times 10^{-2}$) ↓ | MSE ($\times 10^{-3}$) ↓ | SSIM ↑ |
|---|---|---|---|---|
| DDIM Baseline | 25.04 | 9.14 | 4.43 | 0.77 |
| w/o inverse transformation | **27.64** | **5.13** | **2.45** | **0.81** |
| w/ inverse transformation | **27.64** | 5.14 | **2.45** | **0.81** |

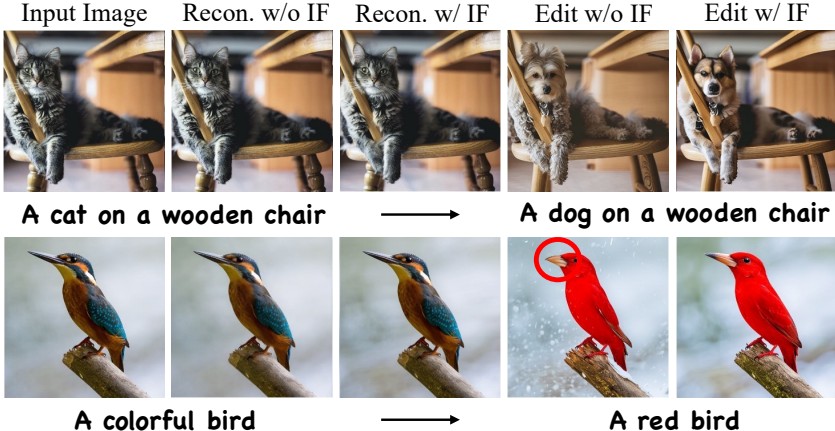

Figure 8: Visualization of the reconstruction and editing results with and without inverse transformation (IF). We adopt PnP as the editing method.

## B   Combination with NTI

To further demonstrate the free-lunch benefit, we supply extensive experiments by combining FreeInv with the previous representative method NTI [25] to further boost performance. The result is provided in Tab. 8.

Table 8: Ablation study on plugging FreeInv into NTI [25].

| Methods | PSNR ↑ | LPIPS ($\times 10^{-2}$) ↓ | MSE ($\times 10^{-3}$) ↓ | SSIM ↑ |
|---|---|---|---|---|
| DDIM Baseline | 25.04 | 9.14 | 4.43 | 0.77 |
| NTI | 26.74 | 5.46 | 3.13 | 0.79 |
| FreeInv | 27.69 | 5.14 | 2.45 | 0.81 |
| NTI+FreeInv | **28.15** | **4.89** | **2.31** | **0.81** |

## C    Experiments on SDXL

Following previous works [25, 45, 15, 50] in the literature, we choose SD1.5 to perform U-Net based experiments, and FLUX to perform DiT based experiments. In Tab. 9, we further include SDXL [30] for comparison with U-Net-based architectures. FreeInv consistently delivers performance improvements, further demonstrating its effectiveness and generality.

Table 9: Ablation study on plugging FreeInv into SDXL [30].

| Methods | PSNR ↑ | LPIPS ($\times 10^{-2}$) ↓ | MSE ($\times 10^{-3}$) ↓ | SSIM ↑ |
|---|---|---|---|---|
| SDXL DDIM Baseline | 24.78 | 12.3 | 6.02 | 0.75 |
| SDXL FreeInv | **26.68** | **5.57** | **3.15** | **0.79** |

## D    Human Evaluation

In Tab. 3, we compare the editing results from PnP [42], MasaCtrl [4], and P2P [12] under the scenarios with or without FreeInv. We use fifteen images from the PIE benchmark, with each editing method applied to five distinct images. During the survey, participants are shown the source image, the edited results with and without FreeInv, and the target prompt. They are then asked to choose the edited image that demonstrates higher textual alignment and better source preservation. Finally, we receive 165 votes from a participant pool. A screenshot of the survey interface is provided in Fig. 9.

In a similar way, we perform another user study to evaluate the effectiveness of different inversion methods. Participants are presented with P2P editing results generated using each inversion method, and then asked to choose their preferred results. Finally, we receive 130 votes from a participant pool, and the result is provided in Tab. 10.

Table 10: User study on different inversion methods.

| User Study | DDIM | NTI | EDICT | DI | VI | PI | Ours |
|---|---|---|---|---|---|---|---|
| preference (%) | 4.6 | 12.3 | 8.5 | 10.8 | 7.7 | 26.2 | **30.0** |

## E    More Visualizations

### E.1    Image Reconstruction

We additionally visualize more reconstruction results in Fig. 10, to compare FreeInv with other state-of-the-art inversion methods. The results further verify the effectiveness and generality of FreeInv as indicated in Sec. 4.2.

### E.2    Image Editing

**Comparison with Other Inversion Methods.** We show more examples edited by P2P with previous state-of-the-art inversion approaches in Fig. 11. The visualization results further validate that FreeInv significantly outperforms the DDIM baseline, and achieves performance comparable to existing state-of-the-art inversion approaches.

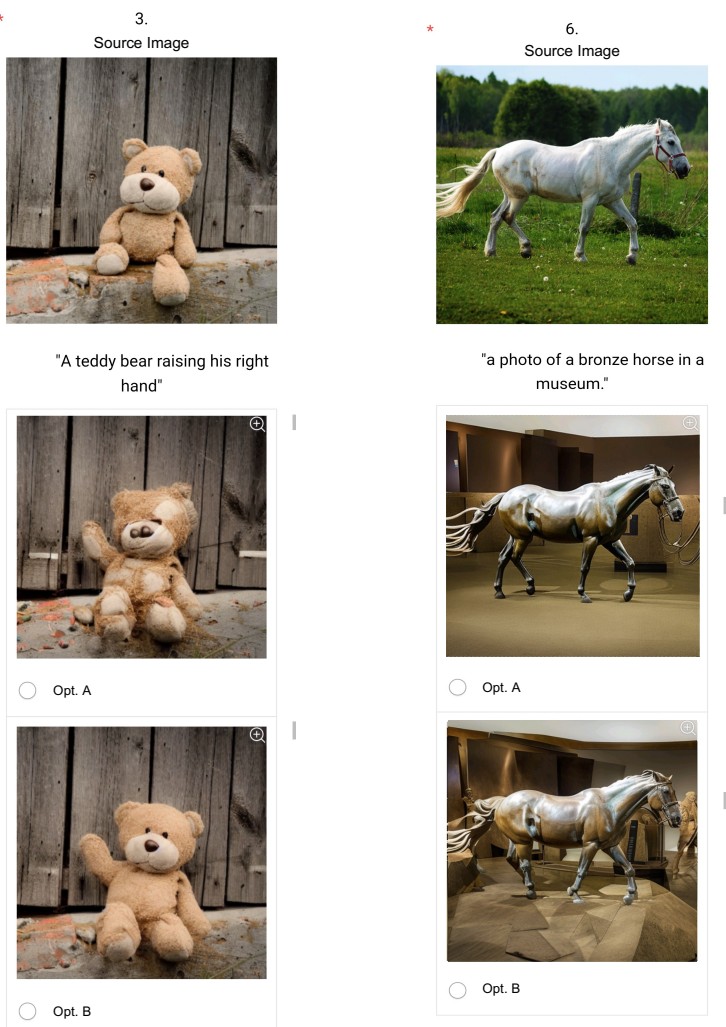

Figure 9: The screenshot of the user survey interface on the phone. The source image, the edited results with and without FreeInv, and the target prompt are presented to the participants.

**Plugging in Existing Image Editing Methods.** Due to operational simplicity, FreeInv can be readily plugged into existing inversion-based image editing frameworks. In Fig. 12 and Fig. 13, we present more editing results of PnP [42] and MasaCtrl [4], respectively. As shown in Fig. 12, 13, FreeInv outperforms the baseline editing method remarkably.

### E.3 Video Editing

We provide the video editing results through this link, where we compare the results of TokenFlow [10] baseline, TokenFlow with STEM-Inv [20], and TokenFlow with FreeInv. Besides, we provide the video reconstruction results of DDIM inversion, STEM-Inv and FreeInv. Through the visualization results, we observe that FreeInv exhibits superior reconstruction fidelity and editing effects, compared with DDIM inversion and STEM-Inv.

## F    Social Impact and Limitation

**Social Impact.** We introduce a free-lunch DDIM inversion-enhanced technique FreeInv in this work. FreeInv enables high-fidelity reconstruction, benefiting image and video editing accordingly. However, we are aware that it can be potentially abused by those malicious individuals or groups

| Input Image | DDIM | NTI | EDICT | BELM | RF-Solver | SD+Ours | FLUX+Ours |
|---|---|---|---|---|---|---|---|

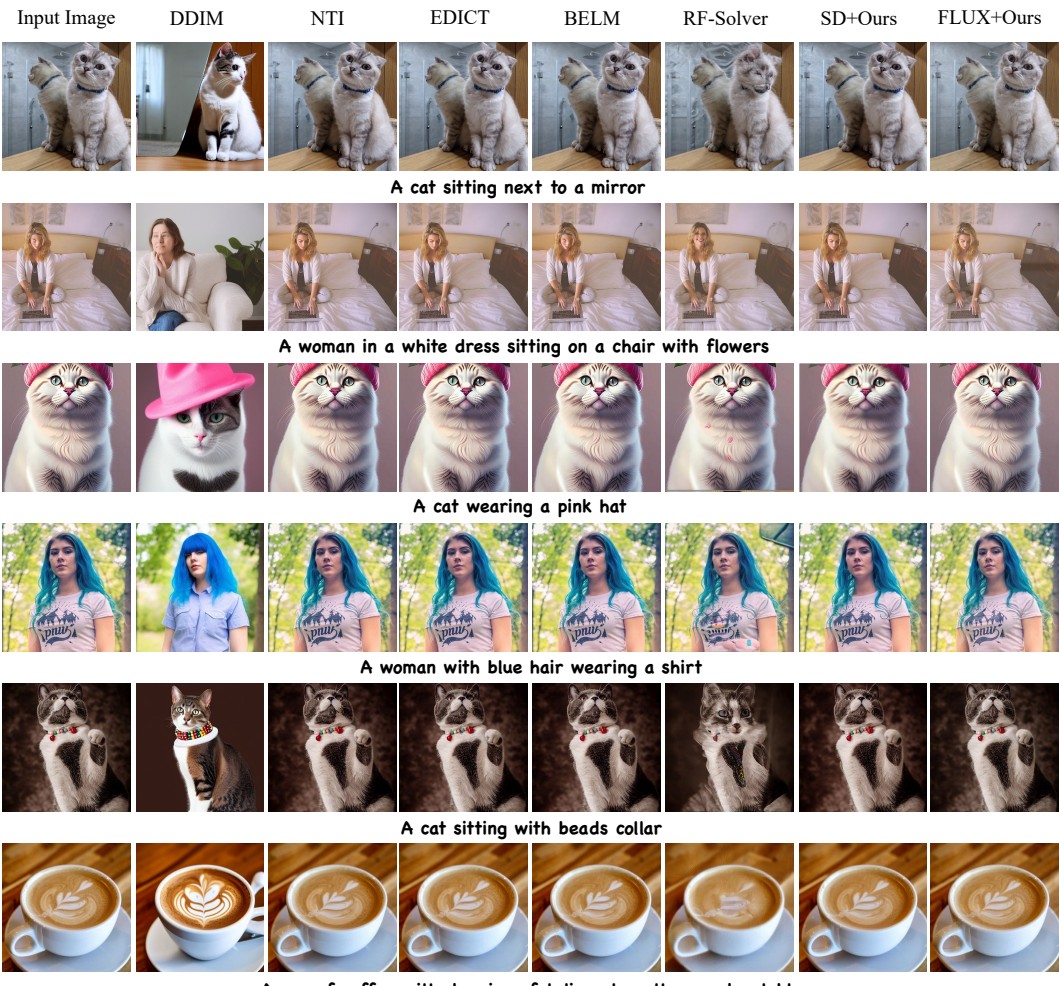

A cat sitting next to a mirror

A woman in a white dress sitting on a chair with flowers

A cat wearing a pink hat

A woman with blue hair wearing a shirt

A cat sitting with beads collar

A cup of coffee with drawing of tulip put on the wooden table

Figure 10: **Qualitative comparison.** Visualization of the reconstruction results in comparison with state-of-the-art inversion methods.

to distribute false information and cause confusion, which undoubtedly violates the intention of our research. We believe the misuse can be alleviated through developing AIGC detection algorithms and being supervised with regulations.

**Limitation.** While our proposed FreeInv improves the efficiency of DDIM inversion significantly, there still remains room for improvement. One key challenge lies in balancing editability and reconstruction fidelity, which is a common issue for inversion methods. In some cases, FreeInv may overemphasize the preservation of source content, which can limit its editability. Another limitation lies in that FreeInv is an inversion-enhanced technique. The editing quality also relies on the editing framework itself which is integrated with FreeInv. Thus, a better editing framework may yield better editing result. For specific scenarios, a suitable editing framework should also be considered.

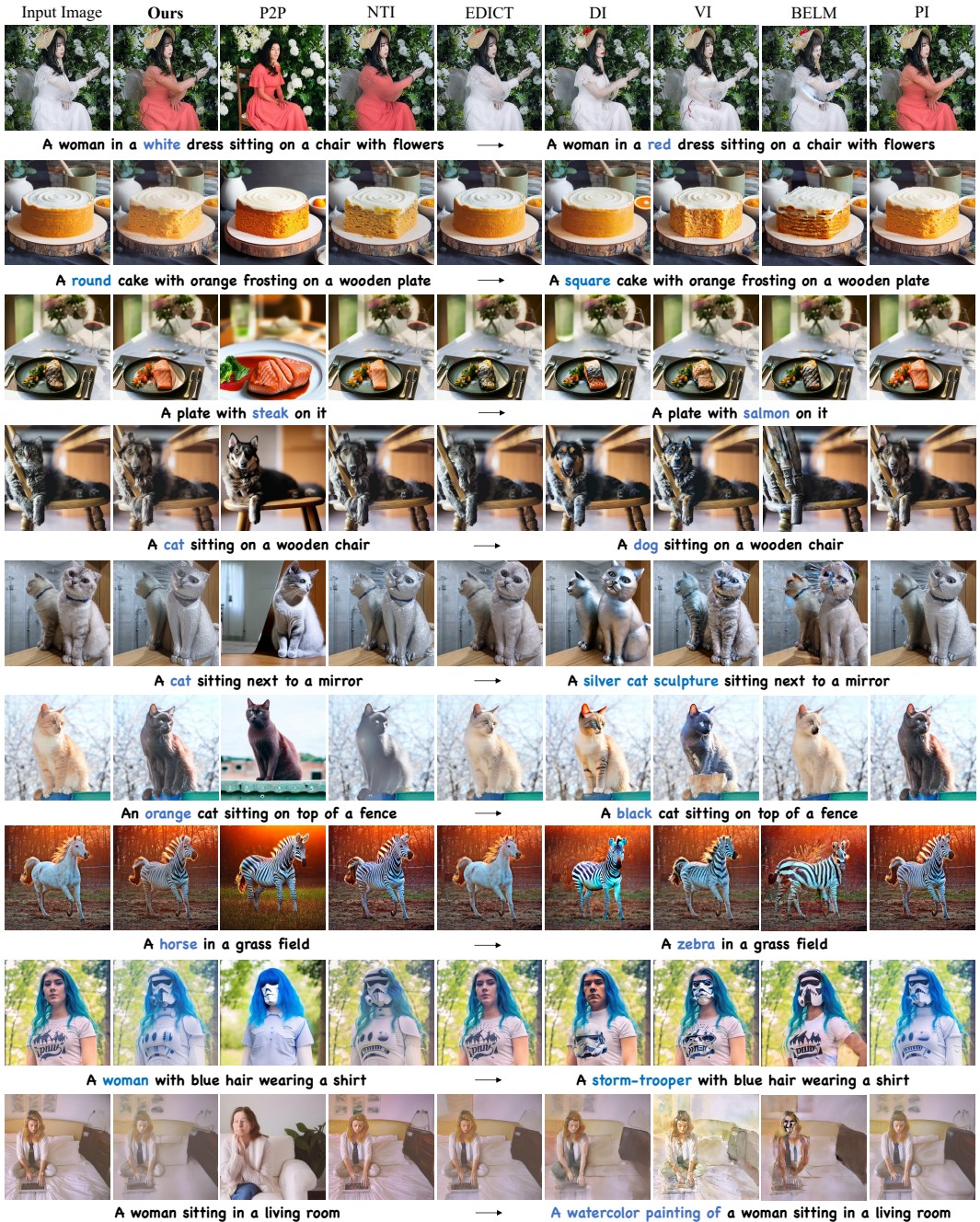

Figure 11: **Qualitative comparison.** More comparison with state-of-the-art inversion methods. P2P [12] with DDIM inversion serves as the baseline method, and all of the methods are plugged into it.

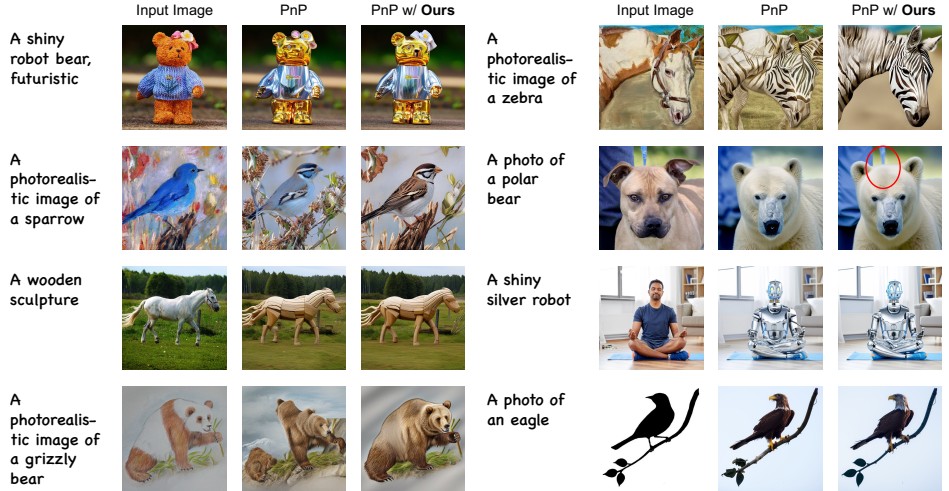

Figure 12: **Qualitative comparison.** More editing results of PnP [42] with and without FreeInv.

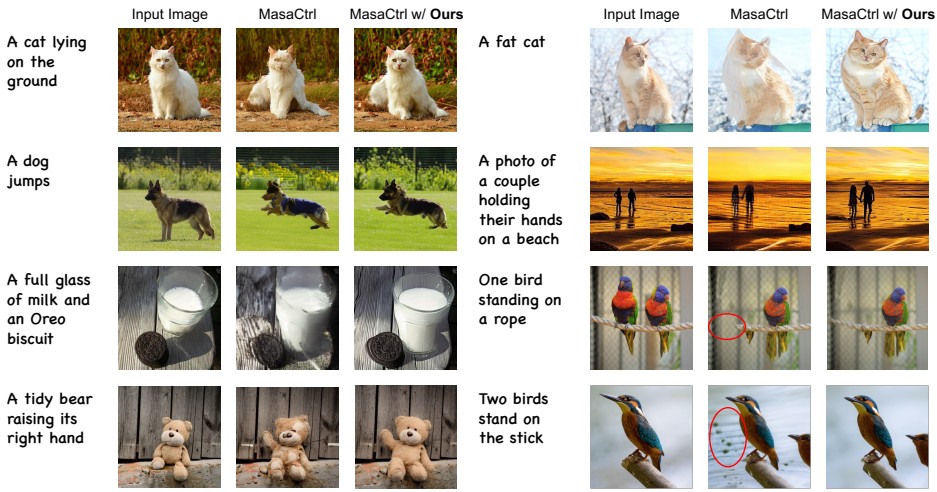

Figure 13: **Qualitative comparison.** More editing results of MasaCtrl [4] with and without FreeInv.

