# OpenReview forum: "FreeInv: Free Lunch for Improving DDIM Inversion"
_NeurIPS.cc/2025/Conference — NeurIPS 2025 poster_

### Official Review · Reviewer_rX7b · 2025-06-25

**Clarity:** 3
**Significance:** 2
**Originality:** 2
**Rating:** 4
**Confidence:** 4

**Summary:**

This paper introduces FreeInv, a method to mitigate inversion errors in DDIM. In standard DDIM inversion, cumulative errors degrade reconstruction and editing quality, and existing solutions often incur substantial computational and memory overhead. FreeInv creates a multi-branch ensemble effect by using one-time MC sampling combined with random image transformations, enabling efficient inversion and reconstruction. The authors demonstrate that FreeInv is adaptable to both image and video data, as well as compatible with various editing methods and model architectures.

**Questions:**

- In Figure 3 and Table 3, the comparisons are limited. In Figure 3, it would be valuable to include other advanced methods (e.g., NTI, PI) that can also be integrated with various editing frameworks. For the user study in Table 3, comparing against other inversion methods would better demonstrate the strengths of the proposed approach.

- The setup for MB-I in Table 6 is unclear. I understand that MB-R uses four branches generated by rotating the input image. What constitutes “independently sampled images” in MB-I? Does this imply using additional distinct images for inversion? Clarification on this point would help readers better understand and interpret the results.

- Have the authors tried using strong augmentations, such as color jitter or contrast adjustments? While these may risk semantic distortion and potentially degrade the ensemble effect, if semantic changes can be controlled, stronger augmentations could increase trajectory diversity and improve performance. It would be helpful to know whether the authors have explored this experimentally.

*Minor Suggestions*
- In Eq. (2), it appears that “t+1” should be replaced with “t,” since the inversion step goes from t to t+1, and the model $ \epsilon_{\theta}$ is applied at timestep t. Accordingly, all subsequent equations should be revised to reflect this correction.

**Ethical Concerns:**

["NO or VERY MINOR ethics concerns only"]

**Final Justification:**

I acknowledge that the theoretical foundation of the proposed method is weak. However, its simplicity and competitive performance compared to existing approaches, without incurring additional computational costs, are its strengths. Additionally, the introduction of the multi-branch concept in the inversion process is a meaningful contribution. Therefore, I raise my score.

**Limitations:**

yes

**Paper Formatting Concerns:**

No concerns

**Quality:**

2

**Strengths And Weaknesses:**

**[Strengths]**
- FreeInv achieves an ensemble effect by applying random transformations, avoiding substantial computational and memory overhead.
- The method is applicable to both image and video data and is compatible with various editing methods and model architectures.

**[Weaknesses]**
- Quantitative performance is not superior to the comparison methods.
- Qualitative results are similar to or, in some cases, slightly worse than those of the comparison methods.

---

> ### Author Rebuttal · Authors · 2025-07-31
>
> Thank you for the constructive comments.
>
> **Q1: Quantitative performance is not superior to the comparison methods.**
>
> **A1:**
>
> (1) As demonstrated in Tab. 1-2, no single method consistently outperforms all the others across all metrics, highlighting the inherent trade-offs in current approaches. Taking a full consideration of the reconstruction and editing performance, FreeInv appears balanced and competitive among the previous methods.
>
> (2) The main advantage of FreeInv is its simplicity and efficiency. Moreover, it can be seamlessly integrated with U-Net and DiT based methods for both image and video inversion, while introducing ignorable memory and computational cost.
>
> **Q2: Qualitative results are similar to or, in some cases, slightly worse than those of the comparison methods.**
>
> **A2:**
>
> (1) In this paper, we aim to show that **without additional computational and memory cost**, FreeInv can achieve competitive results compared with previous methods, such as NTI and PI. We present more visualization results in the supplementary material and project page, to demonstrate FreeInv's effectiveness. We hope the reviewer may refer to those results for more comprehensive evaluation of our method.
>
> (2) To further validate the qualitative results of FreeInv, we supplement a user study among different inversion methods in A3 to Q3. The results also verify the visualization quality of our method.
>
> **Q3: In Figure 3 and Table 3, the comparisons are limited. In Figure 3, it would be valuable to include other advanced methods (e.g., NTI, PI) that can also be integrated with various editing frameworks. For the user study in Table 3, comparing against other inversion methods would better demonstrate the strengths of the proposed approach.**
>
> **A3:**
>
> (1) Sorry for making you confused. We have two dimensions to verify the effectiveness of our method. One dimension is to show our method can be combined with various editing methods. In Fig.3, we show that FreeInv can be seamlessly combined with different inversion-based editing methods, including P2P, MasaCtrl, and PnP. The other dimension is to show our method is competitive against other types of inversion method when integrated within the same editing framework. In Fig. 4, we compare the editing results with different inversion methods (e.g., NTI, PI) , with P2P as the common editing method to ensure fairness.
>
> (2) Thank you for your suggestion, we make a more comprehensive user study that consists of the editing results with different inversion methods, where each participant is asked to choose the most visually appealing and prompt aligned editing results. Finally, we receive 130 votes from a participant pool.
>
> | User Study    | DDIM | NTI  | EDICT | DI   | VI   | PI   | Ours |
> | ------------- | ---- | ---- | ----- | ---- | ---- | ---- | ---- |
> | preference(%) | 4.6  | 12.3 | 8.5   | 10.8 | 7.7  | 26.2 | 30.0 |
>
> **Q4: The setup for MB-I in Table 6 is unclear. I understand that MB-R uses four branches generated by rotating the input image. What constitutes “independently sampled images” in MB-I? Does this imply using additional distinct images for inversion? Clarification on this point would help readers better understand and interpret the results.**
>
> **A4:** Yes, MB-I means using additional distinct images for inversion. We will clarify this point in our final version.
>
> **Q5: Have the authors tried using strong augmentations, such as color jitter or contrast adjustments? While these may risk semantic distortion and potentially degrade the ensemble effect, if semantic changes can be controlled, stronger augmentations could increase trajectory diversity and improve performance. It would be helpful to know whether the authors have explored this experimentally.**
>
> **A5:**
>
> (1) FreeInv does not impose any spatial or semantic constraints on different branches. As demonstrated in Tab. 6, MB-I constituted with distinct images can bring higher improvements compared to MB-R, for the reason that distinct images enrich the diversity of the inversion trajectories.
>
> (2) During the rebuttal phase, we add the experiment applying color jitter, where we alter the brightness, contrast, and brightness of the original image. The results show that jitter can also serves as an alternative.
>
> |    Methods    | PSNR  | LPIPS($\times10^-2$) | MSE($\times10^-3$) | SSIM |
> | :-----------: | :---: | :------------------: | :----------------: | :--: |
> |     DDIM      | 25.04 |         9.14         |        4.43        | 0.77 |
> |     flip      | 27.47 |         5.43         |        2.55        | 0.80 |
> | patch shuffle | 27.61 |         5.18         |        2.55        | 0.80 |
> |   rotation    | 27.64 |         5.14         |        2.45        | 0.81 |
> |    jitter     | 27.53 |         5.40         |        2.55        | 0.80 |
>
> **Q6: In Eq. (2), it appears that “t+1” should be replaced with “t,” since the inversion step goes from t to t+1, and the model is applied at timestep t. Accordingly, all subsequent equations should be revised to reflect this correction.**
>
> **A6:** In Eq. (2) we follow the officially released code of [8,38], and use $t+1$ rather than $t$.

---

> > ### Comment · Reviewer_rX7b · 2025-08-05
> >
> > Thank you for the authors’ thorough response. I have also reviewed the comments from other reviewers and agree that the theoretical foundation of the proposed method is relatively weak. Nevertheless, its simplicity and lack of additional computational overhead result in performance that is competitive with existing approaches. Furthermore, introducing the multi-branch concept to the inversion process is a meaningful contribution. Accordingly, I will raise my score.

---

### Official Review · Reviewer_WToP · 2025-06-30

**Clarity:** 3
**Significance:** 2
**Originality:** 2
**Rating:** 4
**Confidence:** 4

**Summary:**

This paper presents a DDIM inversion method without optimization (training/fine-tuning), which uses latent representation transformation to obtain multiple add noise/de-noising trajectories to further reduce the mismatch error. The paper includes a theoretical analysis to make the proposed method more trackable. The paper provides some good visual results in image/video reconstruction and editing tasks.

**Questions:**

1. Why is there not much difference between these transformations in table 5? Is the combination an equal probability selection of flip, patch shuffle and rotation?
2.  Why is the A white cat below figure 3 missing a leg even after using FreeInv?

**Ethical Concerns:**

["NO or VERY MINOR ethics concerns only"]

**Final Justification:**

Most of my concerns have been addressed, and I am keeping my initial score.

**Limitations:**

yes

**Quality:**

3

**Strengths And Weaknesses:**

**Strengths.**

1. The paper has a clear motivation, and there are significant speed and storage advantages over methods that require optimizing additional embedding or storing errors in these processes.
2.  The paper presents a theoretical analysis process that helps to understand the rationale and motivation of the approach.
3.  The paper visualizes an intermediate cross- attention map to show a better word-to-pixel alignment than DDIM.
4.  Based on some T2I models and Video generation models, the method visually shows good reconstruction and editing results.

- - -

**Weaknesses.**

1.  The paper needs deeper statistical analysis, and the results in Tables 1-2 show that the overall performance of the method is not as good as emphasized in the paper in terms of reconstruction and editing.
2.  L103-L104, the paper puts forward a strong dependence, but without any theoretical basis, it seems that the argumentations are somewhat subjective.
3.  L118, ensemble techniques lack the necessary citations
4.  Although the paper puts a lot of visualization results, as well as some pictures on the dataset ablation, but the effectiveness of the method is not convincing enough, if we can see some quantitative results on the video and in-depth analysis would be better.

---

> ### Author Rebuttal · Authors · 2025-07-31
>
> Thank you for the constructive comments.
>
> **Q1: The paper needs deeper statistical analysis, and the results in Tables 1-2 show that the overall performance of the method is not as good as emphasized in the paper in terms of reconstruction and editing.**
>
> **A1:**
>
> (1) As demonstrated in Tab. 1-2, no single method consistently outperforms all others across all the metrics, highlighting the inherent trade-offs in current approaches. Taking a full consideration of the reconstruction and editing performance, FreeInv appears balanced and competitive among the previous methods.
>
> (2) The main advantage of FreeInv is its simplicity and efficiency. It can be seamlessly integrated with U-Net and DiT based methods for both image and video inversion, while introducing ignorable memory and computational cost.
>
> **Q2: L103-L104, the paper puts forward a strong dependence, but without any theoretical basis, it seems that the argumentations are somewhat subjective.**
>
> **A2:** Actually, this perspective is proposed by previous literature [7,14,22].
>
> **Q3: L118, ensemble techniques lack the necessary citations**
>
> **A3:** Thank you for your suggestion. We will add citations in our final version of the paper.
>
> **Q4: Although the paper puts a lot of visualization results, as well as some pictures on the dataset ablation, but the effectiveness of the method is not convincing enough, if we can see some quantitative results on the video and in-depth analysis would be better.**
>
> **A4:** We have included quantitative results in our paper. For the video part, in Fig. 6, we report the time cost, memory occupation, and mean PSNR for each video sequences, listed below the corresponding frames.
>
> **Q5: Why is there not much difference between these transformations in table 5? Is the combination an equal probability selection of flip, patch shuffle and rotation?**
>
> **A5:**
>
> (1) As stated in Sec. 3.3 of our paper, we apply image transformations to form different branches. Empirically, we find different types of transformation do not yield too much difference. We think the reason is that whether to exploit a transformation is more important than which type of transformation to exploit.
>
> (2) Yes, the combination in Tab. 5 refers to an equal probability selection of flip, patch shuffle and rotation. This combined augmentation strategy yields the highest trajectory diversity, and thus obtains the best reconstruction performance observed in Tab. 5.
>
> **Q6: Why is the A white cat below figure 3 missing a leg even after using FreeInv?**
>
> **A6:** We think its left leg is positioned behind the right leg. Since the leg color is also white, the left leg is partially occluded and difficult to distinguish.

---

> > ### Comment · Reviewer_WToP · 2025-08-06
> >
> > I appreciate the authors' response. Having read the comments from other reviewers, I concur that the theoretical analysis in the paper is limited, with some derivations lacking clarity and smoothness. Nonetheless, the method delivers compelling results with a relatively simple design. As such, I decide to retain my previous positive score.

---

### Official Review · Reviewer_ko1D · 2025-07-01

**Clarity:** 3
**Significance:** 3
**Originality:** 3
**Rating:** 4
**Confidence:** 4

**Summary:**

This paper presents a lightweight method (FreeInv) for improving the quality of image and video inversion. FreeInv approximates a multi-trajectory ensemble by randomly transforming (e.g., rotating, transformation) the latent representation at each time step and keeping it consistent during inversion and reconstruction, significantly reducing reconstruction errors with little cost.

**Questions:**

1. The rot and flip don't disturb the original image, but wouldn't the patch-shuffle on the latent?
2. Is it feasible to combine FreeInv with existing optimization-based methods like NTI to further boost performance, or would that negate the "free-lunch" benefit?
3. Why is unet-based diffusion model is only on sd1.5? For example, Renoise have experimented on SDXL. It is feasible to experimented on SD3.

Overall, I am positive about this manuscript. Some deeper theoretical and experimental analysis and discussion is necessary, however the manuscript mostly focuses on ‘how to do’ and ‘how well it works’, and lacks in ‘why it works’.

**Ethical Concerns:**

["NO or VERY MINOR ethics concerns only"]

**Final Justification:**

The authors have addressed most of my concerns, and I am maintaining my score.

**Quality:**

3

**Strengths And Weaknesses:**

Strengths:

[+] Easy to follow.
[+] Nice performance

Major weakness:

1. It is really amazing to achieves better results with such an easy method. However, the discussion on the theoretical part is still too weak, in that the theoretical nature of simulating multipath ensemble through spatial perturbations is underpinned by a lack of mathematical rigor.
2. The given image or video visualization seems too simple. I'm wondering if there might be an upper limit. How does it perform in the reconstruction or editing of more complex images (e.g. more detailed images)?
3. In some tasks (image editing), the expense is actually tolerable, and the effect is primary. This finding is interesting, but on its own, if it can't beat the strongest models now available (or by combining extant methods) or doesn't provide enough theoretical proofs, I think it may be difficult to inspire follow-up research.
4. The proposed method outperforms some of the more costly methods, but these more costly methods may perform better in more complex image inversions, whereas simple yet effective methods are often difficult to scale up. I'd like to hear a discussion of that.

---

> ### Author Rebuttal · Authors · 2025-07-31
>
> Thank you for the constructive comments.
>
> **Q1: It is really amazing to achieves better results with such an easy method. However, the discussion on the theoretical part is still too weak, in that the theoretical nature of simulating multipath ensemble through spatial perturbations is underpinned by a lack of mathematical rigor.**
>
> **A1:**
>
> (1) Note that at different time steps, we apply different transformations, e.g., at different time steps, the rotation angles are randomly sampled. Thus, from a high-level, FreeInv can be interpreted as an implicit ensemble of multiple paths which correspond to different transformations.
>
> (2) More rigorously, we can interpret the connection between multi-branch ensemble and applying image transformations from two aspects, as discussed in Sec. 3.3. Firstly, we mathematically show that multi-branch ensemble can be viewed as one-time MC sampling at each time step, which means at each time step, we randomly sample one branch from distinct images. Then, based on this understanding, we randomly transform an image into different augmented versions to form different images/branches as this operation yields more efficient implementations than performing real data sampling. Its principle is similar to using data augmentations to mimic real data sampling in general deep learning practice.
>
> (3) Eq. (6) for multi-branch ensembe does not impose restrictions on the sampled images for different paths. Thus, it still holds for different transformations of a single image.
>
> (4) In our original paper, we conduct a detailed ablation study, summarized in Tab. 6. The results verify that FreeInv remains effective.
>
> **Q2: How does it perform in the reconstruction or editing of more complex images (e.g. more detailed images)**
>
> **A2:**
>
> (1) In Tab. 1, we conduct a convincing quantitative evaluation on PIE benchmark, which is a public benchmark covering a wide range of natural and artificial contents, following previous studies [14,46] in the area.
>
> (2) We have also showed some visualizations on complex or detailed images. For example, in the "A yellow car" case of Fig. 3, the camera captures a car speeding past, with the trees, buildings, and sculptures in the background slightly blurred due to the car's high speed. We consider this a complex scene due to the combination of fast motion and intricate background elements. Our method successfully reconstructs both the foreground car and the motion blur in the background with high fidelity, enabling precise image editing. We will add more visualizations on complex images in revision.
>
> **Q3: In some tasks (image editing), the expense is actually tolerable, and the effect is primary. This finding is interesting, but on its own, if it can't beat the strongest models now available (or by combining extant methods) or doesn't provide enough theoretical proofs, I think it may be difficult to inspire follow-up research.**
>
> **A3:**
>
> (1) No single previous method obtains state-of-the-art results across all the evaluation metrics. As demonstrated in Tab. 2, although EDICT exhibits strong ability for background preservation, it suffers from poor prompt alignment, as reflected by low CLIP similarity scores. On the contrast, DI excels in prompt following, but its background preservation is not well. Taking a full consideration of the background preservation and CLIP similarity, the performance of FreeInv is balanced and competitive among the previous methods.
>
> (2) The main merit of our method is the superior efficiency. Pevious methods (e.g., NTI, EDICT, BELM), whose reconstruction/editing quality are on par with our method, consumes much more computation load, either in memory or time. Some method (e.g., NTI) even needs additional training.
>
> (3) We believe the efficiency is important. As shown in our paper, high efficiency may benefit the editing of video which consists of many frames. Moreover, it may promote future research to apply diffusion models to other downstream tasks, e.g., classification, segmentation, etc.
>
> **Q4: The proposed method outperforms some of the more costly methods, but these more costly methods may perform better in more complex image inversions, whereas simple yet effective methods are often difficult to scale up. I'd like to hear a discussion of that.**
>
> **A4:** In our opinion, we think the simplicity of our method makes it more scalable and easier to adapt. As shown in our paper, our method can be seamlessly integrated with both U-Net-based models (e.g., SD1.5, SDXL) and DiT-based models (e.g., FLUX), supporting both image and video inversion. Moreover, it can be combined with various reconstruction/editing methods to improve the generation quality further, e.g., combining with the previous optimization-based method NTI to further enhance reconstruction fidelity. To the best of our knowledge, FreeInv is *the first method* to achieve such broad compatibility and flexibility across architectures and modalities.
>
> **Q5: The rot and flip don't disturb the original image, but wouldn't the patch-shuffle on the latent?**
>
> **A5:**
>
> (1) Actually, patch-shuffle spatially disrupts the original image, and we employ it to enhance the diversity of inversion trajectories, serving as a role similar to that of flipping and rotation.
>
> (2) FreeInv does not impose any spatial or semantic constraints on different branches. We think the diversity matters. As shown in Tab. 4, MBDI, formulated with distinct images, still brings performance improvements. By leveraging more diverse branches, MBDI attains superior results.
>
> **Q6: Is it feasible to combine FreeInv with existing optimization-based methods like NTI to further boost performance, or would that negate the "free-lunch" benefit?**
>
> **A6:** Yes, it is feasible to combine FreeInv with NTI to further boost reconstruction fidelity.
>
> |    Methods    | PSNR  | LPIPS($\times 10^{-2}$) | MSE($\times 10^{-3}$) | SSIM |
> | :-----------: | :---: | :------------------: | :----------------: | :--: |
> | DDIM Baseline | 25.04 |         9.14         |        4.43        | 0.77 |
> |      NTI      | 26.74 |         5.46         |        3.13        | 0.79 |
> |    FreeInv    | 27.69 |         5.14         |        2.45        | 0.81 |
> |  NTI+FreeInv  | 28.15 |         4.89         |        2.31        | 0.81 |
>
> **Q7: Why is unet-based diffusion model is only on sd1.5? For example, Renoise have experimented on SDXL. It is feasible to experimented on SD3**
>
> **A7:**
>
> (1) We choose SD1.5 to perform U-Net based experiments, since most of previous studies [22,41,12,46] in the literature utilize this model. Therefore, we also perform experiments with SD1.5 for a fair comparison.
>
> (2) SD3 is a DiT-based model whose architecture is similar to that of FLUX. However, FLUX demonstrates stronger generative capabilities and has become the preferred choice in DiT-related research. Therefore, in this paper, we adopt FLUX for all DiT-based experiments.
>
> (3) To make a more convincing conclusion and consider the limited time during the rebuttal phase, we further include SDXL for comparison with U-Net-based architectures. FreeInv consistently delivers performance improvements, further demonstrating its effectiveness and generalizability.
>
> |    Models    | PSNR  | LPIPS($\times 10^{-2}$) | MSE($\times 10^{-3}$) | SSIM |
> | :----------: | :---: | :------------------: | :----------------: | :--: |
> |     SDXL     | 24.78 |         12.3         |        6.02        | 0.75 |
> | SDXL+FreeInv | 26.68 |         5.57         |        3.15        | 0.79 |

---

> > ### Comment · Reviewer_ko1D · 2025-08-04
> > **Response to authors**
> >
> > Thank you for the detailed rebuttal and for providing additional experiments on SDXL and the combination with NTI. I appreciate the effort to address the reviewers' concerns. The new results are helpful and have clarified some of my initial questions, particularly regarding the method's compatibility and generalizability.
> >
> > However, after careful consideration of your response and the comments from other reviewers, my primary concerns remain, and in some areas, have been reinforced.
> >
> > My main reservations are centered on the theoretical justification for the method.
> >
> > 1.  **On the Theoretical Foundation:** Your rebuttal explains FreeInv by drawing an analogy to a multi-branch ensemble, viewing it as a form of one-time Monte Carlo sampling (A1 to my review, A1 to Reviewer aAd8). While this provides a high-level intuition, it does not, in my opinion, provide the mathematical rigor I was hoping for. The core issue, also strongly highlighted by **Reviewer aAd8**, is the lack of a clear theoretical link between the proposed random spatial transformations and the mitigation of the DDIM mismatch error, which originates from the discrepancy in the ODE solver formulations. The argument feels more like a post-hoc justification based on an analogy to data augmentation, rather than a principled derivation.
> >
> >     *   **Could you provide a more direct mathematical argument? Specifically, how does ensembling via spatial transformations (rotation, flip, etc.) provably reduce the expected value of the mismatch error ε as defined in your paper's Eq. (3)? The connection between a spatial transform and the temporal integration error of the ODE is the missing link that I believe is critical to understanding the method's effectiveness.**
> >
> > 2.  **On the Performance and Contribution:** I acknowledge your new results, including the impressive LPIPS improvement on SDXL. I also accept your point that FreeInv is "balanced and competitive" (A3 to my review). However, the paper is titled "Free Lunch," which sets a high bar. As noted by **Reviewers WTOP and rX7b**, the quantitative performance is not consistently superior to existing, more complex methods. While the efficiency is a clear advantage, the trade-off seems to be a non-trivial drop in performance or quality in some cases, or simply being "on par" in others.
> >
> >     *   **Given that the theoretical underpinnings are not fully solidified, and the empirical performance is competitive but not decisively state-of-the-art, could you elaborate on what you see as the most compelling, fundamental contribution of this work? Is the main takeaway simply that "a form of data augmentation during inversion helps," or is there a deeper principle at play that we are missing?**

---

> > > ### Author Response · Authors · 2025-08-07
> > > **Looking Forward to Your Reply**
> > >
> > > Dear Reviewer ko1D,
> > >
> > > Thank you for your valuable feedback. We hope our response adequately addresses your concerns. As the deadline for the discussion phase approaches, if you have any other questions or would like to discuss further, please let us know. We sincerely look forward to your reply.
> > >
> > > Best regards from all authors

---

> ### Author Response · Authors · 2025-08-04
> **Further Clarification by Authors**
>
> Thank you for providing your valuable feedback. In response to your concern, we provide further clarification below.
>
> **Q8: More direct mathematical argument.  Specifically, how does ensembling via spatial transformations (rotation, flip, etc.) provably reduce the expected value of the mismatch error ε as defined in your paper's Eq. (3)?**
>
> **A8:**
>
> (1) As explained in Eq. (3), the reconstruction error is determined by the mismatch error $|\epsilon_\theta\left(x_{t+1}, t+1\right)-\epsilon_\theta\left(x_t, t+1\right)|$. Therefore, the key is to mitigate the mismatch error.
>
> (2) Through Eq. (6), it can be mathematically proved that the mismatch error of multi-branch inversion is no larger than that of a single branch in each time-step on expectation. The inequality explains why MBDI is able to reduce the mismatch error.
>
> (3) When the different branches are constituted with one image applied different transformation, where $x_t^i$ refers to the latent representation after applying the $i$-th type of transformation, it can be proved that the Eq. (6) still holds. This indicates that the ensemble of multiple transformation yields smaller mismatch error compared to that of a single branch in each time-step on expectation.
>
> (4) On the lefthand of Eq. (6), $\epsilon_{\theta, \tilde{\boldsymbol{\lambda}}}^{e}\left(x_t, t+1\right)$ corresponds to the statistical expectation $E_{\lambda}[\epsilon_{\theta, \lambda}^{e}(x_{t}, t+1)]$ (see Eq. (7,8) for the proof). Therefore, it is reasonable to estimate the expectation using MC sampling (Eq. 9), which provides an unbiased estimate. Although lower variance of the estimation can be obtained through more sampling times, we find a single sample is enough, considering both effectiveness and efficiency (see Tab. 4).
>
> (5) Overall, we randomly sample the $i$-th branch, which corresponds to applying the $i$-th type of transformation to the latent variable in Eq. (10). Based on the above analyses regarding transformations and the use of one-time Monte Carlo sampling, it can be shown that the inequality in Eq. (6) still holds.
>
>
> **Q9: What is the most compelling, fundamental contribution of this work?**
>
> **A9:**
>
> (1) Our method is by no means a mere implementation of data augmentation during inversion; rather, it makes a meaningful contribution to the advancement of the field. Previous works have only considered single-branch DDIM inversion, with methods constrained and limited to memory-based [12,14,22] and numerical-based [41,42,43] techniques. To the best of our knowledge, we are the first to introduce multi-branch DDIM inversion (MBDI) to the literature. We provide both theoretical and empirical evidence that MBDI effectively reduces mismatch error.
>
> (2) However, simply proposing MBDI may result in an ordinary contribution, as MBDI does not inherently offer advantages over existing single-branch methods. Indeed, it also introduces additional computational and memory costs. Building upon MBDI, we further eliminate these extra costs through one-time Monte Carlo sampling and transformation, ultimately leading to the proposal of FreeInv. FreeInv achieves superior reconstruction fidelity compared to NTI (PSNR 27.69 vs. 26.74), while demonstrating significantly higher efficiency in both time (4s vs. 148s) and memory usage (3031 MB vs. 11945 MB).
>
> (3) Not only do we present the community with a novel alternative for improving DDIM inversion, but we also highlight a new perspective on how to enhance inversion performance by introducing additional branches, which has rarely been explored. We hope the paper can inspire further research in this area.
>
>
> We deeply appreciate your time and consideration. If you have any further concerns or queries, please do not hesitate to let us know.

---

### Official Review · Reviewer_aAd8 · 2025-07-01

**Clarity:** 2
**Significance:** 2
**Originality:** 2
**Rating:** 3
**Confidence:** 4

**Summary:**

This paper proposed to adopt extra transformation operation to mitigate the inconsistency issue in the inversion process of FMs and DMs.

**Questions:**

See the weaknesses, please.

**Ethical Concerns:**

["NO or VERY MINOR ethics concerns only"]

**Final Justification:**

The theory part is not good enough, however, its practical usage is recognized.

**Limitations:**

yes.

**Paper Formatting Concerns:**

no.

**Quality:**

1

**Strengths And Weaknesses:**

Strengths:

-	The paper is well written and easy to follow.

Weaknesses:

-	(My main concern) The theoretical derivation in the section 3 did not persuade me to believe that the extra rotation operation in the inversion would result in less reconstruction error. Based on the theory in BELM, the reconstruction inconsistency is introduced by the discrepency of the formulations of DDIM and its inversion. Based on that theory, I cannot understand why rotation will mitigate this. I think a deeper theoretical explanation of FreeInv is needed.

-	The math of (5) and (6) cannot deduce to a final smaller inconsistency error of the inversion path. The average error across different branched should not matter to the inversion of the path. And the process of FreeInv does not really mimic the MBDI, cause it is not a branch, but an image rotating.

---

> ### Author Rebuttal · Authors · 2025-07-31
>
> Thank you for the constructive comments.
>
> **Q1: The theoretical derivation in the section 3 did not persuade me to believe that the extra rotation operation in the inversion would result in less reconstruction error. Based on the theory in BELM, the reconstruction inconsistency is introduced by the discrepency of the formulations of DDIM and its inversion. Based on that theory, I cannot understand why rotation will mitigate this. I think a deeper theoretical explanation of FreeInv is needed.**
>
> **A1:**
>
> (1) From a high-level, our theoretical goals are consistent with BELM. Our derivation also originates from the discrepancy between DDIM and its inversion, which is termed as mismatch error formulated in Eq. (3).
>
> (2) To reduce the mismatch error, BELM interprets DDIM process as solving ordinary differential equations (ODE) and increases the error order by employing a linear multi-step ODE solver, which is a standard technique for improving ODE solution accuracy.
>
> (3) In contrast, we adopt a different approach from BELM. While we still use a first-order solver, we propose ensembling the DDIM trajectories across multiple images, as shown in Eq. (4). We further show in Eq. (6) that the multi-branch version of DDIM can reduce the mismatch error. In Sec. 3.3 we explain how we reduce the computational and memory costs through one-time MC sampling and image transformations.
>
> **Q2: The math of (5) and (6) cannot deduce to a final smaller inconsistency error of the inversion path. The average error across different branches should not matter to the inversion of the path. And the process of FreeInv does not really mimic the MBDI, cause it is not a branch, but an image rotating.**
>
> **A2:**
>
> (1) According to the question, we think the reviewer may misunderstand our operation in MBDI and FreeInv. As discussed in Line 137-138, the righthand of Eq. (6) approximates the expectation of mismatch error of original inversion path without ensemble (note that the $\epsilon$ does not have a superscript $e$). The lefthand of Eq. (6) denotes the mismatch error of the inversion path with ensemble (note that the $\epsilon$ has a superscript $e$). Thus, Eq. (6) actually shows that through multi-branch ensemble, the mismatch error of inversion path can be reduced on expectation.
>
> (2) The key characteristic of different branches in MBDI is that different branches correspond to different images. Thus, we make an approximation in FreeInv, i.e., using different transformations to generate multiple images from a single image. Different transformed images (e.g., images rotated with different angles) in FreeInv correspond to different images in different branches in MBDI. In Sec. 3.3, we have provided detailed theoretical connections between MBDI and our approximation.
>
> (3) In FreeInv, rotation is not the unique choice, and other types of image transformation also work, including flip, patch shuffle, etc., as stated in Line 162 of our paper and proved in our experiments. In Fig. 2 we just take rotation for example to illustrate our method.

---

> > ### Comment · Reviewer_aAd8 · 2025-08-06
> >
> > Thanks for your detailed reply.
> >
> > I still feel confused. Why would other images affect the inconsistency loss of this image in a particular sampling path?
> >
> > I guess for some specific image, you need to rotate it many times, and invert it many times, and rotate back, and calculate an average? This seems like an average for eliminating inconsistency errors and stabilizing the mean? Am I understanding right?

---

> > > ### Author Response · Authors · 2025-08-08
> > > **Looking forward to new feedback**
> > >
> > > Dear Reviewer #aAd8,
> > >
> > > Thank you for your valuable feedback. We have provided clarifications to your new comments. We hope our response can address your concerns. As the deadline for the discussion phase approaches, if you have any other questions or would like to discuss further, please let us know. We sincerely look forward to your feedback.
> > >
> > > Best regards from all authors

---

> > > > ### Comment · Area_Chair_UEM9 · 2025-08-08
> > > >
> > > > Dear Reviewer #aAd8,
> > > >
> > > > Thank you for your efforts in the reviewing and discussion period. Could you please review the authors’ response and let us know whether your concerns have been addressed?
> > > >
> > > > Thank you, The AC

---

> ### Author Response · Authors · 2025-08-07
> **Further Clarifications by Authors**
>
> Thanks for the reviewer’s valuable feedback.
>
> **Q3: Why would other images affect the inconsistency loss of this image in a particular sampling path?**
>
> **A3:** The inversion/reconstruction of different branches is not performed independently. As shown in Eq. (4) and discussed in Line 122-125, **at each time step**, the inversion/reconstruction is performed with ensembled noise (i.e., the mean of noise predictions from different sampling paths or branches) rather than the noise predicted by a single branch. This interaction **at each time step** makes other images affect the mismatch error of current image.
>
> **Q4: I guess for some specific image, you need to rotate it many times, and invert it many times, and rotate back, and calculate an average? This seems like an average for eliminating inconsistency errors and stabilizing the mean? Am I understanding right?**
>
> **A4:** From the reviewer’s comments, we think there still exists misunderstanding. We make further clarifications below.
>
> (1) In MBDI, the interaction of different branches occurs at each time step. That is, at each time step, the inversion/reconstruction of each branch is performed with noise predictions ensembled from all the branches. This means that we do not perform inversion/reconstruction independently for each branch and then average them. Instead, we average the noise predictions from different branches at each time step.
>
> (2) FreeInv is a simplified version of MBDI. In our previous response, we have clarified the connection between image transformation and the branch in MBDI. In FreeInv, at different time steps, we apply different transformations, e.g., at different time steps, the rotation angles are randomly sampled. From a high-level, FreeInv can be interpreted as an implicit ensemble of multiple paths which correspond to different transformations.
>
> (3) More rigorously, we can interpret the connection between multi-branch ensemble and applying image transformations from two aspects, as discussed in Sec. 3.3. Firstly, we mathematically show that one-time MC sampling at each time step (i.e.,at each time step, randomly sampling one branch from distinct images) is an unbiased estimation of multi-branch ensemble. Then, based on this understanding, at different time steps, we randomly transform an image into different augmented versions to form different images/branches as this operation yields more efficient implementations.
>
> (4) As clarified in previous response, Eq. (6) shows that through multi-branch ensemble, the mismatch error of inversion path can be reduced on expectation. As Eq. (6) does not impose restrictions on the sampled images for different paths, it still holds for different transformations of a single image.
>
> We hope the explanations provided above clarify our approach and contribute to a fair assessment. We are glad to engage in further discussion if you have any questions.

---

> > ### Comment · Reviewer_aAd8 · 2025-08-08
> >
> > Thanks for your clarification.
> >
> > One more question please. For an inversion of an image, at the timestep of conducting FreeInv, do we need extra NFEs for  different augmented versions?

---

> > > ### Author Response · Authors · 2025-08-08
> > > **Further Clarifications by Authors**
> > >
> > > Thanks for the reviewer’s valuable feedback.
> > >
> > > **Q5: For an inversion of an image, at the timestep of conducting FreeInv, do we need extra NFEs for different augmented versions?**
> > >
> > > **A5:** No, FreeInv does not require additional NFEs. It only needs a single forward pass through the U-Net or DiT at each time step, as detailed in Eq. (10) and Line 165-172.
> > >
> > > We hope the explanations above can clarify our approach. We welcome any further discussion if the reviewer has any questions.

---

> > > > ### Comment · Reviewer_aAd8 · 2025-08-09
> > > >
> > > > Okay, I somehow understand the intuition.
> > > >
> > > > I will raise the score to 3. I think polishing the presentation of the theoretical intuition could enhance its rigor.
> > > >
> > > > Given the practical utility, if other reviewers believe this paper merits acceptance, I am comfortable with their opinion.

---

> > > ### Author Response · Authors · 2025-08-09
> > > **Looking forward to new feedback**
> > >
> > > Dear Reviewer #aAd8,
> > >
> > > Thank you for your time and consideration. Given the limited time remaining for discussion, we would be grateful for any final questions or comments at your earliest convenience. We sincerely look forward to your reply.
> > >
> > > Best regards from all authors

---

### Comment · Area_Chair_UEM9 · 2025-08-02
**Reviewer-Author Discussion Period**

Dear Reviewers and Authors,

Thank you for your efforts during the review and rebuttal phases. We are now in the discussion period, open until August 6, 11:59 PM AoE.

This paper has received mixed ratings. The authors have provided detailed rebuttals to each review. At your earliest convenience, please kindly read the rebuttals and respond to engage in the discussion. In particular, please indicate whether your main concerns have been addressed.

Thank you again for your contributions.

Best regards,
The AC

---

### Note · Authors · 2025-08-12

Dear reviewers and AC,

We sincerely appreciate the reviewers’ time and consideration, as well as their valuable feedback. In this work, we propose FreeInv, a simple yet effective method to mitigate the trajectory deviation issue in DDIM inversion in a nearly free-lunch manner. The simplicity, efficiency and compelling results of our approach have been recognized as key strengths and meaningful contributions by all reviewers.

Initially, the reviewers raised some questions about the theoretical understanding of our method. During the rebuttal and discussion phase, we summarized the theoretical analysis presented in our original paper and clarified our perspective accordingly. We believe the theoretical analysis and further clarifications are sufficient to answer the reviewers’ questions regarding our method. We will carefully revise the manuscript according to the reviewers‘ constructive comments.

Overall, we believe our work makes a solid contribution to the field. Benefiting from its simple yet effective design, FreeInv is expected to support and inspire future research in this area. We sincerely appreciate the reviewers’ feedback and hope the Area Chair will provide a fair assessment of our work.

Best regards from all authors

---

### Decision · Program_Chairs · 2025-09-17

**Decision:**

Accept (poster)

**Comment:**

This paper presents a lightweight method, called FreeInv, for improving the quality of image and video inversion. FreeInv approximates a multi-trajectory ensemble by randomly transforming the latent representation at each time step and keeping it consistent during inversion and reconstruction, significantly reducing reconstruction errors with little cost. All reviewers acknowledge the simplicity and practical effectiveness of the proposed method. Their main concern is that the theoretical analysis of the method is not sufficiently rigorous. The authors’ response addressed most of the reviewers' concerns, resulting in a final set of ratings: 3 "Borderline Accept" and 1 "Borderline Reject". The reviewer who assigned the "Borderline Reject" rating (aAd8) noted they would be comfortable if the paper were accepted. After considering the paper, the reviews, and the rebuttal, the Area Chair recognizes that this paper provides a novel method for improving DDIM inversion, which is simple, clean, and effective. In all, the Area Chair believe that this work makes a valuable contribution to the field and meets the standards of NeurIPS. Accordingly, the Area Chair recommends acceptance. The authors are requested to revise the paper by addressing the reviewers’ comments in the camera-ready version.